# ASYNCHRONOUS RLHF: FASTER AND MORE EFFICIENT OFF-POLICY RL FOR LANGUAGE MODELS

**Michael Noukhovitch**[*] ❋ ⊞  **Shengyi Huang** ✚  **Sophie Xhonneux** ❋ ⊞  **Arian Hosseini** ❋ ⊞
**Rishabh Agarwal** ◉ ❋  **Aaron Courville** ❋ ⊞ ❋

❋ Mila Quebec AI Institute
⊞ Université de Montréal
✚ Allen Institute for AI
◉ Google Deepmind
❋ Canada CIFAR AI Chair

## ABSTRACT

The dominant paradigm for RLHF is *online* and *on-policy* RL: synchronously generating from the large language model (LLM) policy, labelling with a reward model, and learning using feedback on the LLM's own outputs. While performant, this paradigm is computationally inefficient. Inspired by classical deep RL literature, we propose separating generation and learning in RLHF. This enables asynchronous generation of new samples while simultaneously training on old samples, leading to faster training and more compute-optimal scaling. However, asynchronous training relies on an underexplored regime, online but *off-policy* RLHF: learning on samples from previous iterations of our model which give a worse training signal. We tackle the fundamental challenge in this regime: how much off-policyness can we tolerate for asynchronous training to speed up learning but maintain performance? Among several RLHF algorithms we test, online DPO is found to be most robust to off-policy data, and robustness increases with the scale of the policy model. We study further compute optimizations for asynchronous RLHF but find that they come at a performance cost, giving rise to a trade-off. We verify the scalability of asynchronous RLHF by training a general-purpose chatbot from LLaMA 3.1 8B on an instruction-following task ∼40% faster than a synchronous run while matching final performance. Finally, we extend our results to math and reasoning to demonstrate asynchronous RL can finetune Rho 1B on GSM8k ∼70% faster while matching synchronous accuracy.

## 1 INTRODUCTION

Reinforcement learning (RL) is critical for training AI assistants based on large language models (LLMs) to ensure they follow instructions (OpenAI, 2022), are helpful and harmless (Bai et al., 2022a), and are factually accurate (Roit et al., 2023). As LLMs have increased in size and capability, the scale and complexity of RL finetuning for LLMs has also substantially increased. State-of-the-art LLMs are often finetuned for weeks (Llama Team, 2024; Google Deepmind, 2024), presumably with large amounts of compute resources.

Yet the dominant paradigm for RL finetuning of LLMs, online on-policy RL (Ouyang et al., 2022), is computationally inefficient. *Online* RL methods generate a batch of responses from the model, get feedback on this batch (e.g. from a reward model), and update *on-policy* with feedback on exactly this model's responses, before generating the next batch. Recent *offline* methods efficiently learn directly from a fixed dataset of responses and feedback (Rafailov et al., 2023) but they underperform online methods (Xu et al., 2024). Since feedback on a model's own generations is crucial to good performance (Tang et al., 2024a), we propose generating responses *online* but learning *off-policy* on previous iterations' feedback. By running both processes asynchronously and leveraging new efficient generation libraries (Kwon et al., 2023), we can greatly reduce compute time.

---

[*]michael.noukhovitch@umontreal.ca, code at github.com/mnoukhov/async_rlhf

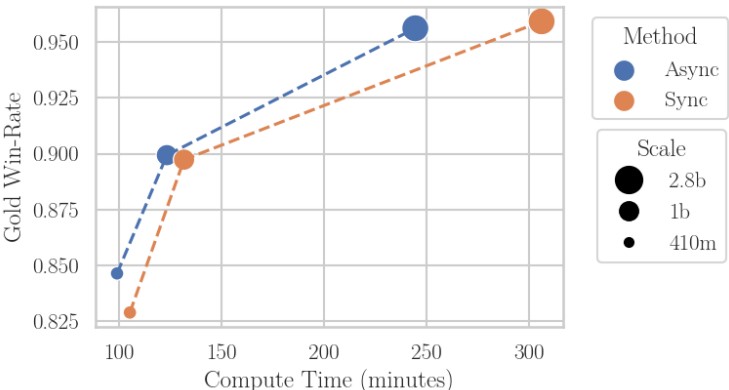

Figure 1: **Asynchronous off-policy RLHF is more computationally efficient**, and matches the win-rate of synchronous on-policy RLHF on TLDR across model scales. On 4×A100 GPUs, it results in training a 2.8B Pythia model 25% faster and improvements in speed increase with scale.

This work focuses on RL finetuning with human feedback (RLHF) and makes a first step into efficient, asynchronous RLHF. We demonstrate strong results and find insights on the widely-used RLHF benchmark, TLDR summarization (Stiennon et al., 2020)

1. We propose asynchronous RLHF and demonstrate that it requires off-policy learning, an underexplored direction for RLHF research. Moreover, we show that RLHF performance generally degrades with more off-policyness.

2. We evaluate many popular RLHF losses and find that Online DPO is most robust to off-policy data and robustness improves with the size of the policy model.

3. We scale model sizes and show that asynchronous RLHF training speed scales better than synchronous RLHF. We achieve the same performance as synchronous state-of-the-art methods ∼ 25% faster with 2.8B models (Figure 1).

4. We demonstrate ways to further optimize compute efficiency in generation-constrained and training-constrained scenarios. In our setup, we improve further and achieve nearly the same performance ∼ 250% faster with 2.8B models.

We then scale up and train a general purpose chatbot by finetuning LLaMA 3.1 8B on a high-quality dataset of human-written demonstrations, No Robots (Rajani et al., 2023)

5. At scale, asynchronous RLHF trains ∼ 40% faster than a synchronous approach and achieves equal performance as measured by GPT-4.

Finally, we demonstrate our results extend to general RL for math and reasoning by finetuning Rho 1B (Lin et al., 2024) on Grade School Math problems (Cobbe et al., 2021)

6. On math, asynchronous RL trains 68% faster than synchronous while matching or exceeding state-of-the-art finetuning numbers (Kazemnejad et al., 2024)

## 2 BACKGROUND

### 2.1 REINFORCEMENT LEARNING FROM HUMAN FEEDBACK

RLHF is a method to align models with hard-to-quantify human preferences using human or synthetic feedback (Christiano et al., 2017; Bai et al., 2022b). In the standard setup for LLMs (Ziegler et al., 2019; Stiennon et al., 2020; Ouyang et al., 2022), we first gather a dataset of prompts $x$ and two responses $y, y'$ (e.g. from our model) and have humans judge which response is better and which is worse. Next, we learn a reward model $r_\phi(x, y)$ on the dataset to approximate human judgement of responses. Finally, we train our model by learning online: iteratively generating responses

to prompts, labelling responses with the reward model, and using RL to optimize the reward. As LLMs are initialized from pretrained weights, RLHF seeks to optimize the reward while maintaining pretrained model abilities. We add a Kullback-Lieber divergence (KL) loss to the objective to keep the model $\pi_\theta$ close to the initial model $\pi_{\text{init}}$ in order to reduce reward model overoptimization (Gao et al., 2022) and alignment tax (Askell et al., 2021).

$$\max_{\pi_\theta} \mathbb{E}_{y \sim \pi_\theta(x)} \left[ r(x, y) - \beta \text{KL}[\pi_\theta(y|x) \| \pi_{\text{init}}(y|x)] \right]$$

The standard method for this approach is Proximal Policy Optimization (PPO; Schulman et al., 2015) which uses an actor-critic framework to optimize the objective. REINFORCE Leave-One-Out (RLOO; Ahmadian et al., 2024) simplifies PPO by reducing to the REINFORCE estimator (Williams, 1992) and empirically estimating a baseline using multiple samples instead of using a value network. Recently Guo et al. (2024); Calandriello et al. (2024) find competitive performance with Online DPO on the RLHF objective. They sample two online continuations, rank them as better ($y_+$) and worse ($y_-$) with the reward model, and optimize the objective of direct preference optimization (DPO; Rafailov et al., 2023).

$$\max_{\pi_\theta} \mathbb{E}_{y_+, y_- \sim \pi_\theta(x)} \left[ \log \sigma \left( \beta \log \frac{\pi_\theta(y_+|x)}{\pi_{\text{init}}(y_+|x)} - \beta \log \frac{\pi_\theta(y_-|x)}{\pi_{\text{init}}(y_-|x)} \right) \right]$$

## 2.2 ASYNCHRONOUS DEEP RL

Prior work in deep reinforcement learning (DRL) has focused mostly on multi-step environments that run on CPU (Bellemare et al., 2013; Tassa et al., 2018; Lillicrap et al., 2019). These algorithms are typically on-policy, meaning the training data comes from rolling out the latest policy. This makes the training synchronous: the learner updates can only occur after policy rollouts, which is slow and can under-utilize hardware resources such as GPUs. To improve throughput and scalability, methods were proposed to parallelize the actor's and learner's computation (Mnih et al., 2016; Espeholt et al., 2018; Berner et al., 2019). Learners and actors can run faster independently but this introduces off-policy data, that is, the rollout data comes from slightly outdated policies. Despite the benefits of asynchronous DRL, to our knowledge, published RLHF works are always synchronous and asynchronous RLHF is severely under-explored.

## 2.3 EFFICIENT LLM TRAINING AND GENERATION

As LLMs have become a more mature technology, a significant effort has focused on improving the efficiency and speed of LLM training and inference. Although some techniques can be leveraged for both (e.g. FlashAttention (Dao et al., 2022)), the problem of efficient training and generation are quite separate and require different methods (Liu et al., 2024). Efficient LLM training involves sharding large models, reducing optimizer states, pipeline batching, and speeding up backpropagation (Rasley et al., 2020; Rajbhandari et al., 2020). Efficient LLM generation focuses custom kernels, effective management of the KV cache, continuous batching (Kwon et al., 2023), and speculative decoding (Cai et al., 2024). As methods have advanced, the backends have diverged and current state-of-the-art libraries for LLM training are separate from LLM inference.

## 3 ASYNCHRONOUS OFF-POLICY RLHF

**On-policy RLHF is Computationally Inefficient** The dominant paradigm for RLHF is fully on-line, on-policy RL: synchronously generate samples then train on these samples using a reward signal (Figure 2, top). To do so, we either (1) use the training library models for both training and inefficient generation, or (2) have generation and training GPUs alternate with some GPUs being idle while the others are working.[1] The second option is clearly inefficient. However, the first option

---

[1]A naive approach is to include both training and generation representations of a model on each GPU but given ever larger LLMs, this isn't feasible memory-wise. A more advanced approach can interleave training and generation backends (Mei et al., 2024; Shen et al., 2024) to utilize both tools. But this incurs overhead from either slow switching between backends or complex manual conversion the two. It also comes at the cost of reduced available memory since the latest inference tools build/optimize execution graphs that must stay

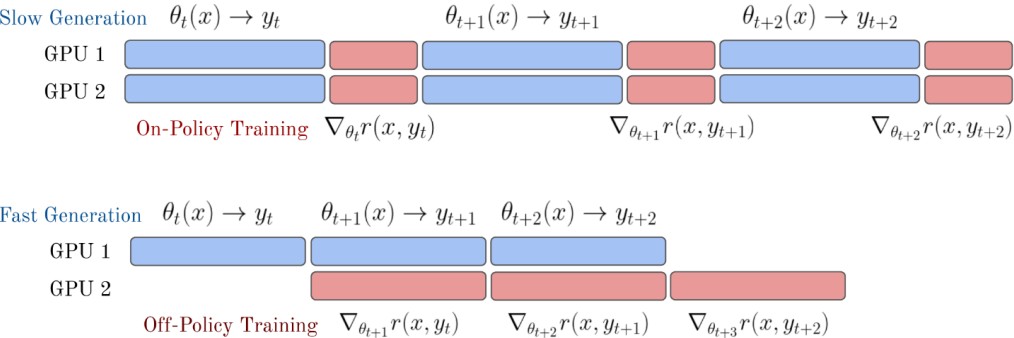

Figure 2: **Synchronous vs Asynchronous RLHF**. **Top:** The current RLHF paradigm synchronously generates and then trains, leveraging the same GPUs for both. This means using slow training libraries for LLM generation. **Bottom:** We propose Cleanba-style (Huang et al., 2023) asynchronous RLHF, separating generation and training to different GPUs. This allows leveraging LLM inference libraries e.g. vllm (Kwon et al., 2023), to greatly reduce generation time. Training time increases because we are learning on only one GPU but the overall runtime for three updates is lower. The caveat is that asynchronous learning requires *off-policy* training: learning on data created by our model at a previous timestep e.g. $\theta_{t+1}$ is updated using data generated by $\theta_t$

does not take into account the divergence between efficient LLM training and generation strategies, as discussed in §2.3. Although training libraries can be used for inference, they are woefully outmatched. For example, let's compare the most popular library for training, Hugging Face transformers (Wolf et al., 2020), with a popular library for inference, vllm (Kwon et al., 2023). We find that vllm is $12\times$ faster than transformers at generating 1024 batches of a modest 128 tokens with a 7B model. Empirically, this gap increases superlinearly with model size. Overall, neither option for synchronous on-policy training is attractive.

## 3.1  OFF-POLICY RLHF

To optimize compute efficiency, it is crucial to separate generation and training on separate GPUs, so each may take full advantage of their optimizations. The clear solution is to use both generation and training GPUs simultaneously and asynchronously. As shown in Figure 2, this requires training on samples that were already generated by our model at a previous iteration, also known as *off-policy* RL. See Appendix D for pseudocode. First, we investigate how off-policy learning affects RLHF methods and then we apply our learnings to optimize compute efficiency for asynchronous RLHF.

**Empirical Setup**   We experiment on the widely-used RLHF benchmark, TLDR Summarization (Stiennon et al., 2020), which provides an SFT dataset of Reddit posts with summaries (Völske et al., 2017) and a feedback dataset of paired summaries where one is rated higher by humans. We follow Gao et al. (2022); Tang et al. (2024a) to create a controlled TLDR setup where we can accurately measure improvements on preferences as well as reward model overoptimization. We relabel the feedback dataset using a well-trained 6.7B "gold" reward model from Huang et al. (2024) so that it acts as a ground truth labeller for our task. Following Huang et al. (2024), we finetune Pythia 410m (Biderman et al., 2023) on the SFT dataset to produce SFT policies and, from the SFT checkpoint, train a reward model on the relabelled dataset. Finally, we train an RLHF policy from the SFT checkpoint using the fixed reward model. We run all methods with a mini-batch size of 512 for 256 steps, so approximately 130,000 samples or "episodes" are seen over the course of training.

**Evaluation**   At inference time, we evaluate success by the win rate, according to our gold model, of generated summaries over the human-written summaries in the SFT dataset. To evaluate alignment tax, we measure how far our RLHF policy has drifted from its SFT initialization using an

---

in GPU memory. Fundamentally, we can do much better optimization and leverage more existing tools for training and inference if they are put on separate GPUs. See Appendix C for a discussion

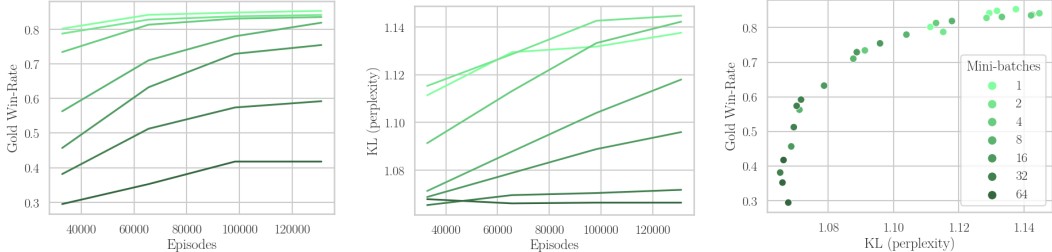

Figure 3: **Trade-off between Win-Rate and KL in Off-Policy PPO**. PPO performance decreases as learning becomes more off-policy. Win-rate is highest when learning is fully on-policy (generate then train on $N = 1$ mini-batches). As we increase $N$, our model must take more steps on data generated by the same old policy. This increases off-policyness and reduces win-rate. **Left:** Gold win-rate over training **Middle**: KL (perplexity) over training, higher is further from initial model **Right:** Gold win-rate vs KL

approximation of the Kullback-Lieber divergance (KL), we measure the SFT model's perplexity on the RLHF policy's summaries.

## 3.2 OFF-POLICY WIN-RATE AND KL

To evaluate robustness to off-policy data, we modify the on-policy RLHF setup to incorporate varying levels of off-policyness. Whereas the on-policy setup generates one mini-batch, labels with reward model, and updates, we propose to generate $N$ mini-batches. Each iteration therefore consists of $N$ mini-batch updates. The first update is fully on-policy as the model has not changed from generation time. But after each mini-batch update and gradient step, the model moves further away from the policy that generated the data. By increasing $N$, we can increase the level of off-policyness of the updates. This setting can correspond to iterative RLHF approaches that generate and label batches of data, e.g. LLaMA 3.1 (Llama Team, 2024).

First, we show the performance of the standard online baseline, PPO, as learning becomes more off-policy. We vary $N$ from 1 (on-policy) to 64 (very off-policy) and plot the gold win-rate and KL over training in Figure 3 (left and middle). We corroborate prior work (Tang et al., 2024a; Tajwar et al., 2024) and find that very off-policy data (and therefore offline data) is worse than on-policy. We extend those results and also find that on-policyness is proportional to learning success for RLHF, with a logarithmic dropoff such that $N = 1$ and $N = 2$ are quite similar.

To accurately compare methods, we plot win-rate and KL against each other in a pareto curve (Noukhovitch et al., 2023) in Figure 3 (right). We find all values of $N$ conform to the same general curve. For PPO, off-policyness did not change the pareto frontier, the fundamental tradeoff of win-rate vs KL of our method. However, off-policyness seems to slow down how training progresses along the frontier. This is in line with previous results from deep RL where data staleness reduces training speed (OpenAI et al., 2019).

## 3.3 ROBUSTNESS OF RLHF LOSSES TO OFF-POLICYNESS

Next, we investigate which RLHF loss is most robust to off-policyness, potentially allowing more asynchronous training. We compare current popular methods, namely PPO, RLOO[2], and Online DPO across a range of off-policyness ($N = 1, 2, 4, 8, 16$) in Figure 4 (left). Although PPO is best at on-policy RL ($N = 1$), its performance is greatly reduced when moving to off-policy learning, as is RLOO's. Online DPO is clearly the most robust to off-policyness. It is able to achieve a higher win-rate at lower KL for slightly off-policy learning ($N = 4$) and is the only method to achieve any reasonably amount of learning in highly off-policy scenarios ($N = 64$).

Both PPO and RLOO only sample 1 completion per prompt whereas Online DPO samples 2. To disentangle this effect, we also run a simple Best-of-2 baseline (Gao et al., 2022) that samples 2

---

[2]To compare the strongest possible methods, we create a modification to RLOO that is robust to off-policyness, see Appendix B

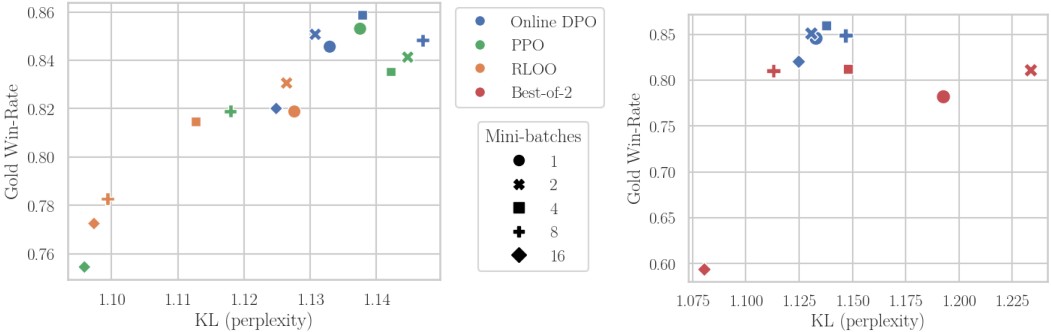

Figure 4: **Robustness of RLHF Losses to Off-Policyness**. Online DPO is more robust to off-policyness than PPO, RLOO (**Left**) or Best-of-2 SFT (**Right**). Performance is shown across levels of off-policyness as mediated by number of mini-batches $N \in \{1, 2, 4, 8, 16\}$. With higher $N$ increasing off-policyness, Online DPO retains much more performance than other methods, as evidenced by off-policy points still being clustered close to optimal performance.

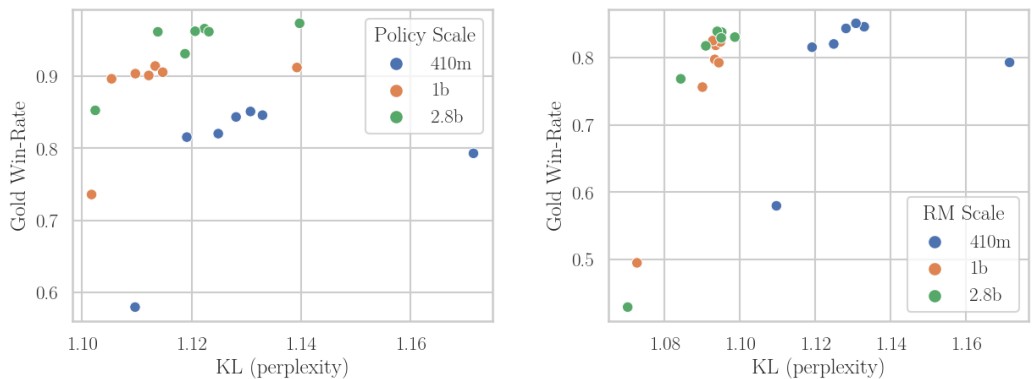

Figure 5: **Scaling Model Size with Off-Policy RLHF**. Plotting the final win-rate vs KL for $N = 1 \rightarrow 64$ mini-batches, covering a spectrum of on-policy to off-policy RL. Scaling policy size (**left**) improves off-policy robustness as seen by tighter clustering of points. But scaling reward model size (**right**) does not, even though it reduces overoptimization, achieving reward with smaller KL.

completions and does supervised finetuning on the completion with the higher reward. We find that Best-of-2 also does not retain performance (Figure 4, right), implying that Online DPO's robustness may be due to the contrastive nature of the loss.

### 3.4  SCALING MODEL SIZE WITH OFF-POLICY RLHF

We scale our setup to Pythia model sizes 410m, 1b, and 2.8b to investigate how scaling affect off-policy RLHF with Online DPO. For clarity, we now plot the *off-policy* pareto curve by taking the final win-rate and KL at each of $N \in \{1, 2, 4, 8, 16, 32, 64\}$. We compare separately scaling the policy and the reward model.

**Scaling Policy**. First, we scale the policy size with a 410m, 1B and 2.8B model while keeping a 410m reward model and show results in Figure 5 (left). As policy size increases, more points on the off-policy pareto frontier are clustered towards the best-performing point. For example, 410m has two points ($N = 16, 32$) far from the optimal area and a wide spread, whereas 2.8b's worst point ($N = 64$) is still quite close to optimal. This means scaling policy size increases robustness: more off-policy runs can approach the best possible win-rate and KL tradeoff.

**Scaling Reward Model**. Next, we scale the reward model across 410m, 1b, and 2.8b while keeping a 410m policy and show results in Figure 5 (right). Following Gao et al. (2022), increasing reward model size allows achieving the same win-rate at a lower KL, reducing overoptimization. Though

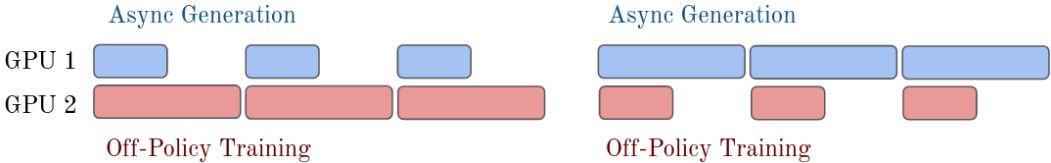

Figure 6: **Asynchronous RLHF can be Training-Bound (left) or Generation-Bound (right)**. In practice, generation and training speeds differ so a challenge of asynchronous learning is how best to balance usage and leverage idle compute time to further improve training.

points are clustering in terms of KL, they are not clustering in terms of gold win-rate. More off-policy points do not achieve relatively better performance, as evidenced by the 410m reward model achieving the highest win-rate for the most off-policy point ($N = 64$). Therefore, we observe that it is only policy scale, not reward model scale, that increases robustness to off-policy learning.

### 3.5 SCALING ASYNCHRONOUS OFF-POLICY RLHF

We apply our learnings to an actual asynchronous RLHF setup. Our results suggest we should aim to be as on-policy as possible so we adapt the simplest, most on-policy asynchronous RL framework, Cleanba (Huang et al., 2023). At time step $t$, we generate completions for prompts with our current model, $y_t \leftarrow \theta_t(x)$, and train on completions generated by our model one timestep back, $\max_\theta r(x, y_{t-1}) + \beta \text{KL}$, as shown in Figure 2. We run both methods on 4 A100 GPUs. For synchronous RLHF, we use all 4 GPUs for both generation and training with Hugging Face transformers. For asynchronous RLHF, we reserve one GPU for generation using the vllm library, and the rest for Online DPO training using Hugging Face transformers. We train the same three scales of model 410m, 1B, and 2.8B and set the policy and reward size to be the same.

Across scales, we find that our one-step off-policy, asynchronous RLHF matches the final win-rate vs KL performance of fully on-policy, synchronous RLHF. In terms of compute, we plot the final gold win-rate against the clock time necessary to reach it in Figure 1. Our method is more efficient at every model size and due to vllm, improvements scale such that at 2.8B, our run is 25% faster.

## 4 OPTIMIZING ASYNCHRONOUS RLHF

Although we have found a significant speedup, the naive asynchronous method is under-utilizing compute. Our model of asynchronous learning requires training and generation to take approximately similar amounts of time, which is not always a reasonable assumption. If the speed of training or generation is mismatched, some of our GPU time will be spent idling, as shown in Figure 6. We propose a solution to take advantage of idling time in each scenario.

### 4.1 GENERATION-BOUND RLHF

Generation and obtaining reward signal can be fundamentally slower than inference. In the classic RLHF setup, generation is autoregressive and scales linearly with the length of the response to generate, whereas reward model inference can be constant. Recent work shows that reward may require human labelling (Llama Team, 2024), output chain-of-thought reasoning (Zhang et al., 2024; Ankner et al., 2024), or executing external tools such as Learn verifiers (Google Deepmind, 2024). In this scenario, we have extra training compute cycles and ask the question, "is it useful to train more on existing data?". Following previous work with PPO (Ouyang et al., 2022), we experiment with taking multiple updates on the same batch of generated data i.e. "ppo epochs" (Schulman et al., 2015). In our asynchronous TLDR setup, we generate $N = 1$ mini-batches and perform $T = 1, 2, 3$ updates per mini-batch.

We plot results across different scales in Figure 7 (left). At 410m and 1B scales, models achieve a higher win-rate for the same number of generated samples, showing that multiple updates make training more sample efficient. This means that extra training time can be used to increase win-rate. But measuring the final points on the pareto frontier in Figure 7 (right), we find that increasing

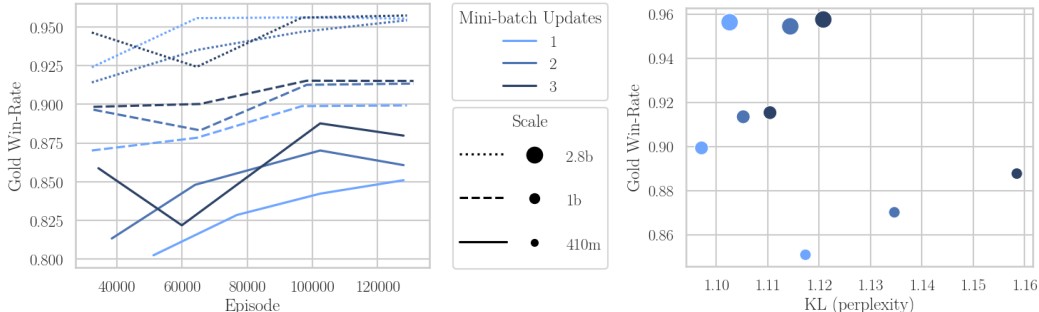

Figure 7: **Optimizing Generation-Bound RLHF**. We can leverage extra training GPU cycles to do multiple updates on the same generated mini-batch ("ppo epochs"). **Left:** At 410m and 1B scales, more updates per batch increases the win-rate achieved at any given episode, making training more data efficient. **Right:** Across scales, more updates change the pareto frontier and cause models to achieve the same win-rate at a higher KL.

updates per mini-batch also increases drift in terms of KL. Therefore, in generation-bound scenarios, multiple updates may increase the win-rate with the same compute-time but incurs higher KL.

## 4.2 TRAINING-BOUND RLHF

The other option is if training is slower than generation. In our 2.8B experiments above, training on 3 GPUs takes twice the time of generating on 1 GPU, so our generation GPU is idling for half the time. We believe that we can sample more continuations to improve Online DPO training. Inspired by the findings of Pace et al. (2024) for reward model training, we propose to generate $K$ samples instead of 2 at each timestep and apply the DPO objective on only on the highest and lowest rewarded completions. In this way, our generation and reward model inference takes $K/2$ times longer while our training remains the same. For TLDR, we experiment with $K = 4$ and find the margin of reward between our highest and lowest samples is approximately $2\times$ larger than our standard $K = 2$ setup. We believe this can provide a more clear gradient for our training and, indeed, find that training proceeds much faster. So we reduce the learning rate $2\times$ and also train for half the number of steps.

We plot the win-rate against compute time across our three scales in Figure 8 (left). We find that we can achieve the same gold win-rate in just over half the time. As we were training-bound, increasing the number of generations, while keeping training samples fixed, did not significantly increase our per-step training time. And $K = 4$ asynchronous training allows us to reduce training steps by half, training $2.5\times$ faster than synchronous. The caveat is that achieving this win-rate comes at a cost of higher KL as shown in Figure 8 (right). Though difference in KL decreases with scale, we still find a visible difference at 2.8B. Similar to generation-bound, optimizing training-bound RLHF can improve speed but at the cost of KL.

# 5 SCALING ASYNCHRONOUS RLHF

## 5.1 GENERAL-PURPOSE CHATBOT

Next, we verify our findings at a larger scale by training an helpful instruction-following chatbot with RLHF. First, we create and label a preference dataset. We finetune LLaMA 3.1 (Llama Team, 2024) on a dataset of 10,000 human-written demonstrations for instructions, No Robots (Rajani et al., 2023) to create our SFT checkpoint. Then, we sample another 3 completion per prompt from our model, to get a total 4 including the human reference in the dataset. We create 6 pairs (4 choose 2) of completions per prompt and use GPT-4o as a judge (Zheng et al., 2023) to create a synthetic preference dataset. We train a reward model on this dataset from the LLaMA 3.1 SFT checkpoint.

We use our best-performing algorithm, Online DPO, and train on 8 H100s sync on-policy and async off-policy for 100,000 episodes. For each sample, we generate a completion of up to 1024 tokens per prompt but, since our model is larger and we generate more tokens, generation using the huggingface

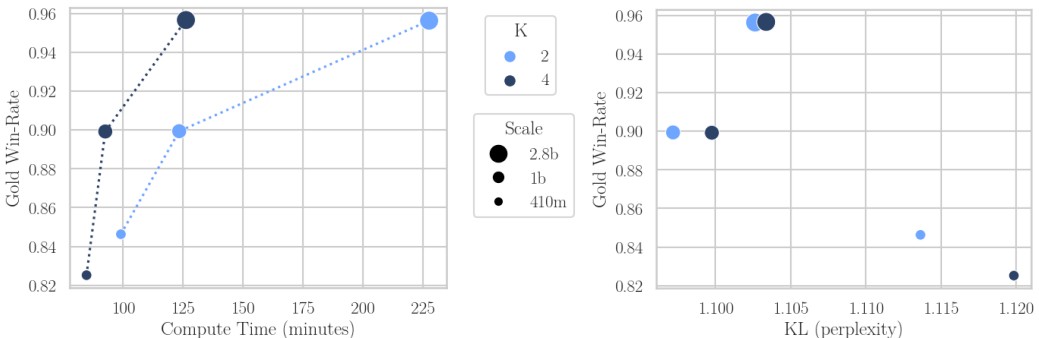

Figure 8: **Optimizing Training-Bound RLHF**. We can leverage extra generation GPU cycles to sample $K$ completions per prompt instead of 2. **Left:** Sampling $K = 4$ improves the gradient such that we can train for half the number of steps and, across scales, achieve the same final win-rate at a fraction of the compute time. **Right:** The trade-off is that increasing $K$ causes models to drift more in terms of KL in order to achieve the same win-rate.

| Model | Win Rate ↑ | Compute Time ↓ | Average Response Sequence Length |
|---|---|---|---|
| SFT | 31.80% | - | 198.40 |
| Sync Online DPO | 57.20% | 230 | 286.21 |
| Async Online DPO | 57.20% | 142 | 290.55 |

Table 1: **Async RLHF Works at Scale for Chatbots.** Async is effective at training a general-purpose chatbot with LLaMA 3.1 8B. It runs 38% faster than sync while matching KL and final GPT4-o win rate against the human-written responses on the No Robots test set (Rajani et al., 2023)

transformers library is too slow to be feasible. So for both sync and async, we reserve one GPU for generation with vllm and the remaining seven for training. Synchronous on-policy learning idles the generation GPU while training and vice versa, whereas asynchronous trains off-policy as previously. We plot the reward and KL over training in Figure 9 in Appendix A.2 and find that async generally achieves the same reward and KL as sync while being 38% faster. We evaluate the final models with GPT-4o as a judge (Zheng et al., 2023), comparing their completions to human-written responses on the No Robots test set. In Table 1, we find async achieves the exact same win-rate as sync, 57.2% while running 38% faster. Overall, we confirm that asynchronous RLHF is equally performant while being faster, even more so at large scale. We note that our async runtime could be even further improved and discuss major considerations in section A.2.

## 5.2 MATH AND REASONING

We now demonstrate that async can work generally for RL with language models using the well-known benchmark of grade-school level math word problems, GSM8k (Cobbe et al., 2021). The setup generally follow Kazemnejad et al. (2024), using Rho-1B (Lin et al., 2024), a state-of-the-art LLM trained on natural language and math corpora, further finetuned on the ground truth reasoning and answers in the training dataset (Havrilla et al., 2024). To train, a reasoning trace and final answer is sampled for each math question, and the reward is set to 1 if the answer string exactly matches ground truth, and 0 otherwise (Singh et al., 2023). We run for approximately 128k prompts and, following Kazemnejad et al. (2024), sample 4 completions per prompt to therefore train for a total of 512k episodes. Final models are evaluated using the pass@1 metric on the test dataset by greedy sampling 1 completion for each question and reporting the percentage of correct answers. Final KL is measured using the perplexity of the base model on the generated completions.

We compare a strong, existing PPO baseline (Kazemnejad et al., 2024) against sync and async Online DPO in a 4 GPU setup as above and report the final evaluations in Table 2. Our sync Online DPO baseline slightly improves over the existing sync PPO baseline. Async, once again, matches sync performance while being more compute efficient, running 68% faster. This large improvement

| Model | Pass@1 on Test Set ↑ | PPL ↓ | Compute Time (Minutes) ↓ |
|---|---|---|---|
| SFT | 40.3% | - | - |
| Sync PPO* | 50.3% | - | 864* |
| Sync Online DPO | **52.2%** | 1.0916 | 218 |
| Async Online DPO | **52.6%** | 1.0922 | **129** |

Table 2: **Async RL works at Scale for Math and Reasoning on GSM8k**. Training Rho 1B on GSM8k, Sync Online DPO outperforms the strong, existing Sync PPO baseline. Furthermore, Async Online DPO is $68\%$ faster and achieves the same performance (pass@1) and KL (PPL) as Sync. *Sync PPO results from Kazemnejad et al. (2024) used comparable 4xA100 GPUs

demonstrates that async RL is perhaps even more suited to reasoning which eschews a reward model and makes efficiency purely about optimizing LLM generation and training.

## 6 RELATED WORK

The most popular attempts at making RLHF more efficient comes in the form of recent offline methods i.e. direct preference optimization (Rafailov et al., 2023, DPO) and followups (Tang et al., 2024b; Rafailov et al., 2024). By directly optimizing a policy using the feedback dataset, their method avoids costly online generation and is much more compute-efficient. But recent works have shown that it is worse than online methods at achieving high reward (Xu et al., 2024) exactly because it eschews online generations (Tang et al., 2024a). Online and, specifically, on-policy data generated by the the model being trained is key to achieving high reward while maintain pretrained model capabilities (Tajwar et al., 2024; Tang et al., 2024b; Agarwal et al., 2023).

Our investigation therefore focuses on optimizing online RLHF methods but not exactly on-policy data. RLHF with off-policy data, generated from previous versions of our model, has been scarcely attempted as no previous methods have focused on asynchronous learning. Munos et al. (2023) provides theoretical arguments for learning from generations by an exponential moving average of the model, however, in practice, Calandriello et al. (2024) finds this to be equal or worse than learning on-policy. Though Tang et al. (2024a) focus on online vs offline methods, one additional experiment in their appendix bears similarities to our $N$ mini-batches setup. Their results imply that more off-policy data decreases online RLHF performance. We greatly extend this direction and investigate which methods perform best off-policy as well as how performance is affected by scale.

This work demonstrates a novel approach to efficiency for RLHF and proposes practical ways to tackle it. Complementary to our work, Mei et al. (2024); Shen et al. (2024) focus on the engineering challenges of efficient, synchronous RLHF and propose clever distributed training techniques to account for generation, reward model inference, and training. Hu et al. (2024) provide another engineering solution that leverages vllm to improve generation speed. Our proposed asynchronous RLHF may remove some of the engineering challenges of synchronous RLHF (e.g. by separating generation and learning), which can make future engineering approaches even more efficient.

## 7 CONCLUSION

This work makes a first step towards asynchronous RLHF, demonstrating how it can improve efficiency while maintaining performance. We demonstrate that an off-policy regime does not have to impact performance and the possibility of further performance/speed tradeoffs. While synchronous RLHF libraries are currently well-optimized and likely outperform our setup, we believe we have proven the viability of asynchronous learning and encourage the community to investigate and optimize this new paradigm. Previously in deep RL, as environments became more complex and model sizes increased, asynchronous learning became the dominant paradigm (Mnih et al., 2016; Berner et al., 2019). In RLHF, model sizes are increasing and recent works have proposed more complex multi-turn environment setups (Shani et al., 2024; Kumar et al., 2024). As such, it seems likely that asynchronous RLHF will become a computational necessity and we believe it is important to turn RLHF research towards this new paradigm and with the challenges it presents.

## REPRODUCIBILITY STATEMENT

We note model training details in Appendix A. Our experiments are based on existing open-source codebases and all code used in the paper is open-sourced on github at `https://github.com/mnoukhov/async_rlhf`. All baseline model checkpoints and training datasets are released on HuggingFace Hub, see github repo for details. To extend this work, one-step async RLHF has been integrated into the `open-instruct` library and notably used for Tulu 3 (Lambert et al., 2025).

## ACKNOWLEDGMENTS

MN thanks Samuel Lavoie for many helpful discussions, Amirhossein Kazmnejad and Milad Aghajohari for great help in GSM8k experiments, MN and AC thank members of Sony Research, Fabien Cardinaux, Lukas Mauch, Stefan Uhlich, James MacGlashan, Bac Nguyen Cong, and Ghouthi Boukli Hacene, for their constant feedback and ideas. MN is funded by Sony, il est aussi soutenu par la bourse Fonds de recherche du Québec - Nature et Technologies. MN is grateful to Mila and ServiceNow for resources used in experiments and grateful to Google for cloud credits provided through a credit award.

## AUTHOR CONTRIBUTIONS

MN led the project, proposed the idea, wrote the code and ran TLDR experiments, and helped write the paper. SH wrote the code and ran large-scale experiments, helped write code for TLDR, proposed Cleanba and wrote key code for asynchronous training, gave feedback in meetings, and helped write the paper. SX wrote code for HH-RLHF that did not end up in the final paper, helped run TLDR experiments, gave feedback in meetings, and helped write the paper. AH wrote some initial code for best-of-n, gave feedback in meetings, and helped edit the paper. RA advised throughout the project, proposed experiments, and helped write the paper. AC was the main advisor, proposed research directions and experiments, and helped edit the paper.

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

# A EXPERIMENT DETAILS

## A.1 TLDR SUMMARIZATION

Experiments on TLDR Summarization are trained using the Hugging Face trl library(von Werra et al., 2023) which leverages Pytorch (Paszke et al., 2019), Accelerate (Gugger et al., 2022), and Datasets (Lhoest et al., 2021). The base models used are the "dedupep" versions of Pythia 410m, 1B, and 2.8B. We follow Huang et al. (2024) for all dataset preprocessing and supervised finetuning hyperparameters. We relabel the dataset with Huang et al. (2024) 6.7B reward model by getting the score for each pair of completions and assigning the completion with the higher score as the "chosen" completion $y_+$, the other being the "rejected" completion $y_-$. We show the baseline results after supervised finetuning, before RLHF training in Table 3.

| Model | Win Rate | KL (Perplexity) |
|---|---|---|
| SFT 410m | 25.36% | 1.075 |
| SFT 1B | 26.82% | 1.071 |
| SFT 2.8B | 35.16% | 1.068 |

Table 3: The win-rate and perplexity of models after supervised finetuning, before RLHF training

For RLHF training, we follow the hyperparameters and suggestions of Huang et al. (2024) with slight modifications. For PPO, see hyperparameters in Table 4.

| Hyperparameter | Value |
|---|---|
| Learning Rate | $3 \times 10^{-6}$ |
| Learning Rate Schedule | Linear |
| Generation Temperature | 0.7 |
| Batch Size (effective) | 512 |
| Max Token Length | 1,024 |
| Max Prompt Token Length | 512 |
| Response Length | 128 |
| Number of PPO Epochs | 1 |
| Total Episodes | 131,072 |
| KL penalty coefficient | 0.05 |
| Penalty Reward Value for Completions Without an EOS Token | -1.0 |

Table 4: PPO Training Hyperparameters

We use the same hyperparameters for all methods with the following method-specific modifications

- RLOO sets $k = 2$
- Online DPO sets $\beta = 0.1$
- Best-of-2 sets learning rate to $1 \times 10^{-6}$ as it tends to overfit quickly

## A.2 NO ROBOTS INSTRUCTION-FOLLOWING

**Hyperparameters** Large-scale experiments were trained with Open Instruct (Wang et al., 2023; Ivison et al., 2023; 2024)[3]. We finetune LLaMA 3.1 (Llama Team, 2024) on a dataset of 10,000 human-written demonstrations for instructions, No Robots (Rajani et al., 2023) to create our SFT checkpoint. The SFT hyperparameters are in Table 5.

Given this SFT checkpoint, we generate a synthetic preference dataset using GPT4-o. First, we generate 3 demonstrations with temperature 0.7 per prompt from the SFT model, totaling 4 generations

---

[3]https://github.com/allenai/open-instruct

| Hyperparameter | Value |
|---|---|
| Model | Meta-Llama-3.1-8B |
| Max Sequence Length | 4,096 |
| Batch Size (effective) | 128 |
| Learning Rate | $5.0 \times 10^{-6}$ |
| Learning Rate Schedule | Linear |
| Learning Rate Warmup Ratio | 0.03 |
| Learning Rate Weight Decay | 0.0 |
| Number of Epochs | 2 |

Table 5: No Robot SFT Model Training Hyperparameters

per prompt when counting the reference completion in the dataset. We create 6 pairs (4 choose 2) of completions per prompt and use GPT-4o as a judge (Zheng et al., 2023) to create a synthetic preference dataset. We train a reward model on this dataset from the LLaMA 3.1 SFT checkpoint, using hyperparameters from Table 6.

| Hyperparameter | Value |
|---|---|
| Model | The Trained No Robot SFT Checkpoint |
| Learning Rate | $3 \times 10^{-6}$ |
| Learning Rate Schedule | Linear |
| Batch Size (effective) | 256 |
| Max Sequence Length | 1,024 |
| Number of Epochs | 1 |

Table 6: Reward Modeling Hyperparameters

Given the SFT model and reward model, we then train Online DPO on 8 H100s synchronously on-policy and asynchronously off-policy for 100,000 episodes. For each sample, we generate a completion of up to 1024 tokens per prompt, an appropriate length for the task. Since our model is larger and we generate more tokens, generation using the huggingface transformers library is considerably slower than vllm (i.e., 20x slower in preliminary testing), and infeasible. So for both sync and async, we reserve one GPU for generation with vllm and the remaining seven for training. Synchronous on-policy learning idles the generation GPU while training and vice versa, whereas asynchronous trains off-policy as previously. Table 7 has the hyperparameters.

| Hyperparameter | Value |
|---|---|
| Model | The Trained No Robot SFT Checkpoint |
| Reward Model | The Trained RM Checkpoint |
| Learning Rate | $8 \times 10^{-7}$ |
| Learning Rate Schedule | Linear |
| Generation Temperature | 0.7 |
| Batch Size (effective) | 256 |
| Max Token Length | 1,024 |
| Max Prompt Token Length | 512 |
| Number of Epochs | 1 |
| Total Episodes | 100,000 |
| Beta (DPO coefficient) | 0.03 |
| Response Length | 1,024 |
| Penalty Reward Value for Completions Without an EOS Token | -10.0 |

Table 7: Online DPO Training Hyperparameters

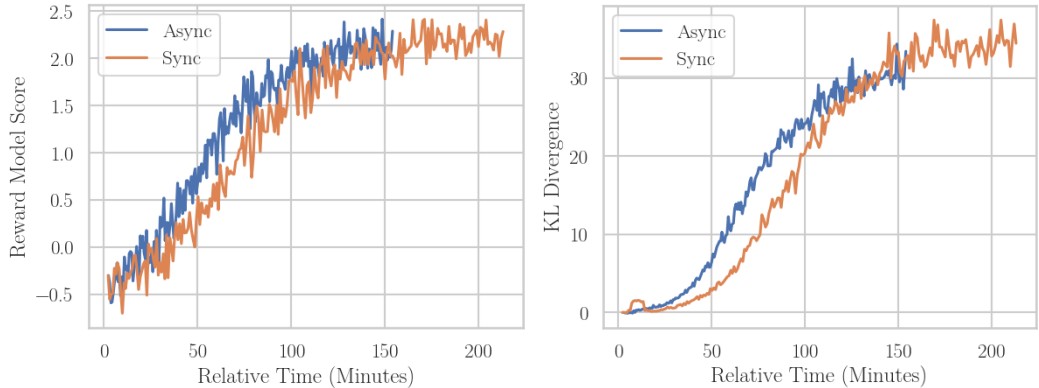

Figure 9: **Asynchronous RLHF works at scale for a General-Purpose Chatbot**. Comparing synchronous and asynchronous online DPO for training an 8B general-purpose chatbot. Asynchronous learning achieves the same reward model score at a lower KL and 38% faster.

For an additional evaluation, we also generate completions on the trained online DPO checkpoints and compare these completions with human-written completions using GPT4-o as a judge. The win rate and average length of generated responses for all models are in Table 8. The async online DPO checkpoint actually obtains exactly the same win rate as the sync online DPO checkpoints. This is perhaps less surprising since both models have very similar KL and scores at the end of the training, as indicated in Figure 9.

| Model | Win Rate | Average Response Sequence Length |
|---|---|---|
| SFT | 31.80% | 198.40 |
| Async Online DPO | 57.20% | 290.55 |
| Sync Online DPO | 57.20% | 286.21 |
| Human | N/A | 179.726 |

Table 8: The trained models' GPT4-o win rate against the human-written responses on the test split of the No Robots dataset (Rajani et al., 2023)

**Practical Considerations for Asynchronous Runtime**    Interestingly, our asynchronous speedup could be even faster. For the synchronous experiments, vllm generation takes 21 seconds and training takes 33 seconds. We have 233 steps of training, so it takes roughly $(21 + 33)$ seconds $* 233 \approx$ 209 minutes. In an ideal setup, we expect asynchronous RLHF to train at the speed of the slower process, training i.e. 33 seconds $* 233 \approx 128$ minutes, roughly 63% faster than the synchronous training time. In practice, though, we find asynchronous training to take 151 minutes: 26 seconds for generation and 39 seconds for training. We note two possible reasons for the slowdown:

1. **Global interpreter lock (GIL)**: With Python, only one thread can execute at any given time and we run a threads for each of generation and training. This issue is mitigated when we call `torch` operations, which can run in parallel internally. However, GIL does occur additional blocking for our generation and learning.

2. **Communication between training and generation**: The generation process must pass generated completions to training and the training process must pass updated model parameters to generation. The latter can be expensive and passing policy parameters is a synchronous GPU call which can slow down training.

Although these issues are outweighed by our improvements, solving them may be important motivation for future work. For example, the latter issue can be mitigated by reducing the frequency of synchronization between generation and learning. One potential solution is generating more mini-batches of data and learning more off-policy as in §3.2.

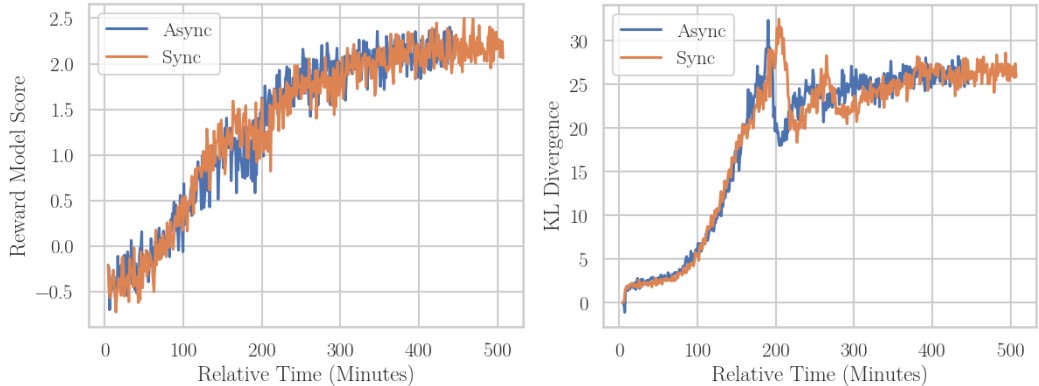

Figure 10: **Asynchronous RLHF also works at scale with PPO**. Comparing sync and async PPO for training an 8B general-purpose chatbot. Async achieves the same reward model score at a similar KL and 38% faster.

| Model | Win Rate ↑ | Average Response Length | Compute Time (Minutes) ↓ |
|---|---|---|---|
| SFT | 31.8% | 198.40 | - |
| Sync Online DPO | **57.2%** | 286.21 | 213.04 |
| Async Online DPO | **57.2%** | 290.55 | **154.03** |
| Sync PPO | 53.0% | 220.35 | 507.33 |
| Async PPO | 52.6% | 229.42 | 446.08 |

Table 9: **Async PPO also matches Sync PPO while being faster for General-Purpose Chatbots**: Trained model GPT4-o win rate against the human-written responses on the No Robots test set (Rajani et al., 2023), average length of the generated responses, and compute time to train on 8xH100 GPUs. Just as with Online DPO, Async PPO closely matches the performance of Sync PPO while being faster to train. Though GPT-4o judges Online DPO to be most performant, PPO models generate notably shorter responses.

**PPO**   We aim to verify that asyncrhonous RLHF will work with other methods at scale as well. We therefore run the same setup as §5 with PPO, instead of Online DPO. All hyperparameters are the same Online DPO, see Table 7, except we decrease the KL coefficient to $\beta = 0.01$ as the original value did not perform well for PPO. We plot the training curves in Figure 10. As previously, we find that asynchronous learning nearly exactly matches the performance of synchronous learning, while being faster. We note a strange spike in KL for both runs, perhaps due to instability of PPO. We evaluate the performance of the final models using GPT-4o win-rate in Table 9 and find that asynchronous PPO nearly exactly matches the performance of synchronous PPO. Overall asynchronous learning is shown to be effective for PPO as well as Online DPO.

Although PPO achieves a similar reward model score to Online DPO, it performed worse when evaluated by GPT-4o. This is likely due to the instability of PPO's optimization and difficulty in finding the best possible hyperparameters. PPO is also more than 2x slower than Online DPO as it requires maintaining a value network in memory which reduces batch size and also training the value network which takes time.

## A.3   GSM8K

**Hyperparameters**   We mainly use the hyperparameters of Kazemnejad et al. (2024) but modify them slightly, as shown in Table 10. Kazemnejad et al. (2024) only experiment with PPO and RLOO (as well as a variant of PPO) where they sample 8 completions per prompt. We found 4 completions to have the same performance as 8 for RLOO so we sample and train on 4 completions. For Online DPO, we sample 4 completions per prompt then choose the best and worst as our DPO pair, as in §4.2. This means that sampling takes the same amount of time as RLOO, but training is faster

| Hyperparameter | Value |
|---|---|
| Model | Rho-1B SFT on GSM8k |
| Learning Rate | $3 \times 10^{-6}$ |
| Learning Rate Schedule | Constant |
| Generation Temperature | 0.7 |
| Max Prompt Token Length | 512 |
| Response Length | 512 |
| Number of PPO Epochs | 1 |
| Batch Size (effective) | 252 |
| Number of Completions per Prompt | 4 |
| Total Prompts Seen | 129024 |
| Total Episodes | 516096 |
| Beta (DPO and KL coefficient) | 0.05 |

Table 10: Online DPO and RLOO Training Hyperparameters for GSM8k

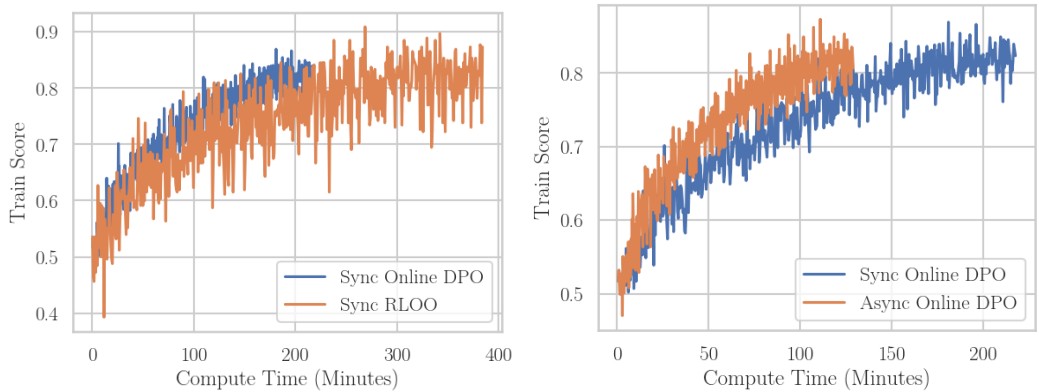

Figure 11: **Online DPO is performant on GSM8k, Async Online DPO is performant and faster**. **Left:** Sync Online DPO matches the general train performance of sync RLOO as measured by the train score over compute. Both methods are run for 512k episodes but Online DPO trains on only the top/bottom of 4 completions, so runs faster. **Right**: Async Online DPO is 68% faster than Sync for GSM8k training and reaches a nearly identical train score.

since we throw out 2 samples, leading to speed improvements seen in Figure 11 left. Preliminary experiments taking the best and worst 2 for RLOO yielded worse results.

Due to the length of outputting reasoning steps, GSM8k requires generating 512 tokens for the output. This makes generation with HuggingFace transformers infeasible[4]. For our 4 GPU experiments, we therefore synchronously generate on one GPU with vLLM and train on the other three with transformers, as in §5, alternating training and generation[5]. We run on 4xL40s GPUs.

**Online DPO outperforms RLOO, PPO** Our base model achieves 40.3% pass@1 on the GSM8k test set. We run RLOO, and Online DPO and use existing numbers from a well-tuned PPO baseline from Kazemnejad et al. (2024). We plot RLOO vs Online DPO train score (percentage of correct answers per batch) in Figure 11 (left) and the final results in Table 11. We find that Online DPO outperforms RLOO and achieves 52.6% final pass@1 after 512k episodes. In comparison, Kazemnejad et al. (2024)'s well-tuned PPO achieves 50.1% after 650k episodes. We also note that our synchronous Online DPO takes $\approx$ 3.5 hours to run on 4xL40s 48Gb GPUs whereas Kazemnejad et al. (2024) synchronous PPO takes $\approx$ 14.4 hours on larger 4xA100 80Gb GPUs with comparable

---

[4]Generating a batch 1024 examples with transformers takes $\approx$ 60 seconds on 4 x 80GB A100 GPUs with all available optimizations like Flash-Attention 2 (Dao, 2023). In contrast, vLLM takes only $\approx$ 11.5 seconds running on a single 80GB A100

[5]This corresponds to the synchronous RLHF paradigm used by Hu et al. (2024)

| Model | Pass@1 on Test Set ↑ | PPL ↓ | Compute Time (Minutes) ↓ |
|---|---|---|---|
| SFT | 40.3% | - | - |
| Sync PPO* | 50.3% | - | 864* |
| Sync RLOO | 50.0% | **1.0778** | 385 |
| Sync Online DPO | **52.2%** | 1.0916 | 218 |

Table 11: **Online DPO is the most performant method for GSM8k**. Final models' pass@1 on the GSM8k test set, a heuristic measure of KL, and compute time to train on 4xL40s GPUs. Online DPO improves over RLOO and a well-tuned PPO baseline while Asynchronous Online DPO achieves the same results 68% faster. *Sync PPO scores and times are from Kazemnejad et al. (2024) trained with comparable 4xA100 GPUs

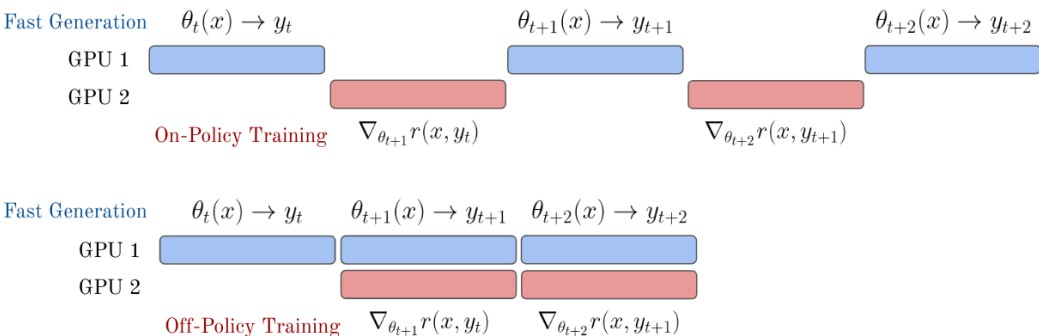

Figure 12: **Comparing Simple Synchronous Training to Asynchronous Training. Top:** The simple but effective approach to efficient synchronous training, e.g. implemented by Hu et al. (2024), separates training and generation onto different GPUs and leverages a state-of-the-art generation library like vLLM to generate and state-of-the-art training library like Deepspeed for training. In order to train synchronous, you idle generation while training and vice-versa. **Bottom:** Asynchronous RLHF speeds up training by training off-policy on previous steps' generations and therefore removes idling time.

speed while also leveraging vLLM for generation and deepspeed for training. This demonstrates the speed and effectiveness of our synchronous baseline. Online DPO also required essentially no hyperparameter tuning to achieve reasonable performance, as opposed to PPO and RLOO. We also note that Kazemnejad et al. (2024)'s proposed method VinePPO, an advanced version of PPO that relies on more samples, outperforms Online DPO with a Pass@1 of 53.4% but it requires much more compute time ($\approx 68$ hours). We do not claim that Online DPO is state-of-the-art for GSM8k but note it is a strong baseline.

**Asynchronous Speedup Analysis** Here, we explain how we achieve the speedup in our GSM8k experiments. We visually demonstrate the synchronous and asynchronous paradigms in Figure 12. As noted above in the details above, this synchronous paradigm is necessary as HuggingFace transformers is too slow for generation so we must leverage vLLLM. We also note this synchronous paradigm is used in an existing competitive library, OpenRLHF (Hu et al., 2024).

In our GSM8k experiments, training takes up 3 GPUs and generation takes 1. In our synchronous setup with Online DPO, generation takes on average 12.2 seconds, getting the reward (evaluating the answer) takes 0.10, and the training step takes 12.8 seconds. This adds up to 25.1 seconds whereas the average actual step time is 25.5 seconds, showing that synchronous training adds an overhead of 0.4 seconds. Asynchronous training runs generation and training at the same time but at the cost of increased overhead. Since we are training-bound, we would expect the average step time to be 12.9 seconds but our actual step time is 15.1 seconds. Although we save a lot of time by running training and generation asynchronously, we lose some speed due to 2.2 seconds in overhead, for reasons outlined in section A.2.

## B  OFF-POLICY RLOO

We wish to use a formulation of RLOO (Ahmadian et al., 2024) that is robust to off-policy data. Flet-Berliac et al. (2024) argue that the formulation is already robust to off-policy data. But both empirically and theoretically, we find this isn't the case. Below, we argue for our off-policy RLOO formulation, which we call Proximal RLOO.

RLOO (Ahmadian et al., 2024) with $k = 2$ samples 2 completions for each prompt from the model $y_1, y_2 \sim \pi_\theta(\cdot|x)$ then updates the loss objective

$$L(\theta) = \frac{1}{2} \left[ \log \pi_\theta(y_1|x) \left( R(y_1, x) - R(y_2, x) \right) - \log \pi_\theta(y_2|x) \left( R(y_2, x) - R(y_1, x) \right) \right]$$

For simplicity, we will focus on the gradient of just one sample $y_1$ and write the baselined reward as an advantage $\hat{A}(y_1|x) = R(y_1, x) - R(y_2, x)$

$$L_{RLOO}(\theta) = \log \pi_\theta(y_1|x) \hat{A}(y_1|x)$$

We can see that RLOO is just REINFORCE with a baseline and the gradient of the loss is quite standard

$$\nabla_\theta L_{RLOO}(\theta) = \nabla_\theta \log \pi_\theta(y_1|x) \hat{A}(y_1|x)$$

Contrastive Policy Gradient (CoPG; Flet-Berliac et al., 2024) proposes an RLHF algorithm that is argued to be robust to off-policy data and has connections to RLOO. In our online, off-policy setup, samples are taken from a previous policy $\pi_{old}$. Here, CoPG can be seen as a modification of RLOO with $k = 2$ divided by the log-probability of the sample under the policy that generated it, $\pi_{old}$.

$$L_{CoPG}(\theta) = \log \frac{\pi_\theta(y_1|x)}{\pi_{old}(y_1|x)} \hat{A}(y_1|x)$$

As shown in Flet-Berliac et al. (2024), this has the exact same gradient as vanilla RLOO

$$\nabla_\theta L_{CoPG}(\theta) = \nabla_\theta \log \frac{\pi_\theta(y_1|x)}{\pi_{old}(y_1|x)} \hat{A}(y_1|x)$$
$$= \nabla_\theta \log \pi_\theta(y_1|x) \hat{A}(y_1|x)$$

This is argued to mean that RLOO is already a good objective for off-policy data but given that there is no reference to $\pi_{old}$, we don't see how this can be the case.

Instead, we leverage an off-policy RLOO that follows the framework and suggestions of Proximal Policy Optimization (PPO; Schulman et al., 2017). Specifically, our loss uses an importance sampling ratio (Sutton & Barto, 2018):

$$L(\theta) = \frac{\pi_\theta(y_1|x)}{\pi_{old}(y_1|x)} \hat{A}(y_1|x)$$

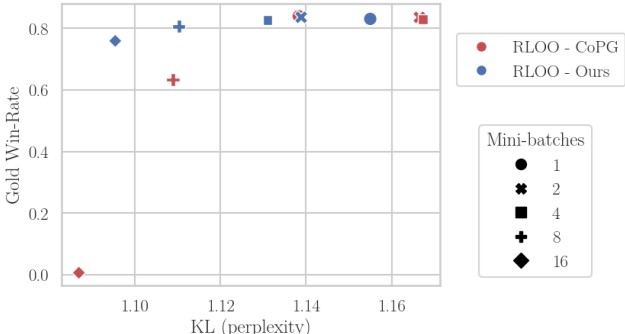

Figure 13: **Our Proximal RLOO outperforms CoPG-style RLOO for online, off-policy learning**

This ratio is still present in the gradient, which we derive with the log-probability trick (Huang et al., 2022; mglss & nbro, 2019):

$$\nabla_\theta L(\theta) = \nabla_\theta \frac{\pi_\theta(y_1|x)}{\pi_{old}(y_1|x)} \hat{A}(y_1|x)$$
$$= \frac{\pi_\theta(y_1|x)}{\pi_\theta(y_1|x)} \nabla_\theta \frac{\pi_\theta(y_1|x)}{\pi_{old}(y_1|x)} \hat{A}(y_1|x)$$
$$= \frac{\pi_\theta(y_1|x)}{\pi_{old}(y_1|x)} \frac{\nabla_\theta \pi_\theta(y_1|x)}{\pi_\theta(y_1|x)} \hat{A}(y_1|x)$$
$$= \frac{\pi_\theta(y_1|x)}{\pi_{old}(y_1|x)} \nabla_\theta \log \pi_\theta(y_1|x) \hat{A}(y_1|x)$$

This demonstrates our loss gives the RLOO gradient with an importance sampling ratio between our current policy and the policy that generated the data $\pi_{old}$.

We also add PPO's clipping of the importance sampling ratio (here renamed $r_\theta$) to within $\epsilon$ of 1, for stability.

$$L_{final} = \min\left(r_\theta(y_1)\hat{A}(y_1|x), \text{clip}(r_\theta(y_1), 1-\epsilon, 1+\epsilon)\hat{A}(y_1|x)\right)$$
$$\text{where } r_\theta(y_1) = \frac{\pi_\theta(y_1|x)}{\pi_{old}(y_1|x)} \tag{1}$$

We call this method, Proximal RLOO, in reference to PPO. We compare the two methods in terms of off-policy robustness using our setup in § 3.3. As shown in Figure 13, CoPG performance drops to 0 as data becomes more off-policy ($N = 16$). In contrast, our PPO-style RLOO remains robust.

## C    WHY EFFICIENT SYNCHRONOUS RLHF IS NOT FEASIBLE

### C.1    TRAINING LIBRARIES ARE INEFFICIENT FOR GENERATION

Whereas asynchronous learning can fully leverage state-of-the-art generation libraries, a naive approach to synchronous learning will generate using the training library (von Werra et al., 2023). We demonstate the necessity of efficient generation libraries by comparing the most popular open-source training library HuggingFace Transformers (Wolf et al., 2020) and a popular generation library vLLM (Kwon et al., 2023) in Figure 14. It is clear that generating with a training library is infeasible at larger scales.

More advanced approaches may attempt to integrate both efficient training and generation into a single backend, e.g. Deepspeed-Chat's Hybrid Engine (Yao et al., 2023). But specific generation libraries, like vLLM, are known to be "substantially better" and lead to large performance gains (Hu et al., 2024).

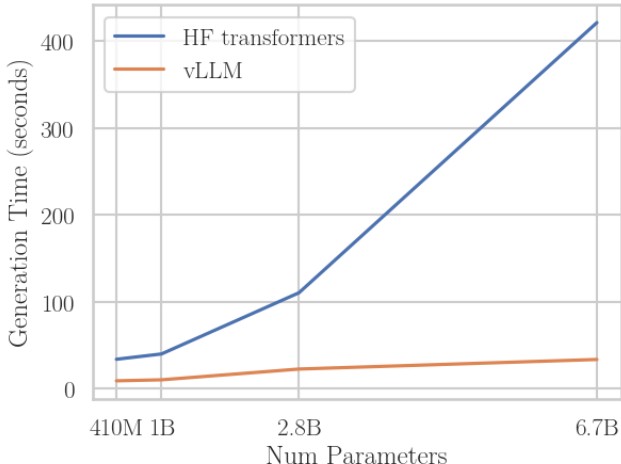

Figure 14: **vLLM is much faster than HF transformers** Comparing the time to generate 128 tokens from a batch of 512 examples of prompt length 512 tokens each. Scaling model sizes from Pythia 410m to 6.7B, we see that vLLM is not just faster at each model scale, the difference is exponential. It becomes infeasible to generate from large models using a training library like HuggingFace transformers

## C.2   INTEGRATING GENERATION INTO SYNCHRONOUS RLHF TRAINING IS DIFFICULT

Since generation libraries are so much more efficient, an intelligent approach to synchronous RLHF must integrate the generation libraries into itself. For a best-case scenario, we consider the arguable state-of-the-art synchronous RLHF library, NeMo-Aligner (Shen et al., 2024).

For NeMo-Aligner's PPO, it combines an efficient training backend, Megatron-LM (Shoeybi et al., 2020), with an efficient generation backend, TensorRT-LLM (NVIDIA, 2024b). In order to leverage both of these, Shen et al. (2024) implement a clever but complex system to convert training models to the generation backend on the fly. Although this feat is done impressively quickly, it still comes with downsides

**Reduced Available Memory, Slower Training**   Building the TensorRT-LLM engine is expensive, so it is better to build it once and keep it in GPU memory. Therefore the training run has less available memory to use. So training must be done with gradient checkpointing to reduce memory usage in backprop, this makes training slower.

**Dynamic Model Resharding, More Overhead in Generation**   Training is done using pipeline parallism to reduce memory but comes the increased cost of overhead communication. In contrast, inference could leverage tensor parallelism to reduce overhead. If there is enough space, models must therefore be re-sharded (which takes time) before converting from training to inference or suffer increased communication overhead. In both cases, there is increased overhead for generation.

## C.3   MAINTAINING GENERATION IN SYNCHRONOUS RLHF IS VERY DIFFICULT

Despite these hurdles, NeMo-Aligner is quite performant . . . for now. The issue is that there are continual updates to both the training backend, Megatron-LM, and the generation backend, TensorRT-LLM. As a case study, we look how NeMo-Aligner is maintained as its underlying libraries change.

NeMo-Aligner was originally built with TensorRT-LLM version 0.11 as its generation backend. By the time of its release on September 8, 2024 TensorRT-LLM had already upgraded to version 0.12 and included new, necessary features like support for the SOTA open-source model LLaMA 3.1 (Llama Team, 2024).

The maintainers of NeMo-Aligner began working to integrate TensorRT-LLM 0.12 into their library (Kong, 2024) but as they were working on it, TensorRT-LLM 0.13 was released. They quickly

adapted the PR and after one and a half months of work, they integrated TensorRT-LLM 0.13 into NeMo-Aligner. The same week, TensorRT-LLM released 0.14.

Each new version of the library brought important speed and feature developments such as LLaMA 3.1 support (0.12), KV cache reuse for LoRA (0.13), and fast logits copying (0.14) as well updating the underlying TensorRT library and fixing important bugs. Despite NeMo-Aligner and TensorRT-LLM both being developed by NVIDIA, it was still infeasible for the NeMo-Tensor team to quickly integrate updates to the generation library.

Generation libraries are generally built as stand-alone libraries (Kwon et al., 2023). Synchronous RLHF must integrate new developments and manually work around any new paradigms, breaking changes, and force those libraries to cooperate in their training paradigm. This makes it infeasible to keep up with the latest developments. In contrast, asynchronous RLHF can use those libraries as stand-alone processes that run parallel to training and integrating new updates is mostly frictionless.

### C.4 Synchronous RLHF is already partially Asynchronous

Although state-of-the-art synchronous RLHF uses the same GPUs for generating and training the policy, it may still leverages asynchronous reward / critic models. NeMo-Aligner's (Shen et al., 2024) PPO training has to leverage four models

- PPO policy (for training and generation)
- reference policy (for KL divergence loss)
- PPO critic (to compute value estimates)
- reward model (to provide reward for completions)

Using PyTriton (NVIDIA, 2024a), the policy and reference policy are on one set of GPUs, but the critic and reward model are actually placed on a completely separate set of GPUs. The two servers (policy and critic/reward model) run and communicate asynchronously to permit pipelining (Shen et al., 2024).

This pipeline can suffer from the same resource allocation issues as noted in §4 so Shen et al. (2024) suggest reserving compute allocation sizes such that [reward model inference + critic inference] ≈ [policy generation + reference policy inference] and [critic train] ≤ [policy train + policy inference initialization].

Therefore, synchronous training libraries may already be partially set up to handle asynchronous training. A fully asynchronous NeMo-Aligner would have to create a third PyTriton server with just the policy for generation and perhaps add another restriction to the compute allocation sizes, a relatively minimal change.

## D   Asynchronous Algorithm

---

**Algorithm 1** Cleanba-style (Huang et al., 2023) Asynchronous RLHF

---

**Initialize:** base model $\pi_\theta$, reward model $R$, dataset $D$, RLHF Loss $L$ (e.g. PPO, Online DPO)

Generate a first batch of completions $y_0 \sim \pi_{\theta_0}(x_0)$
**for** batch of prompts $x_i \in D$ **do**
    send previous prompts $x_{i-1}$ and completions $y_{i-1}$ to TRAIN
    send current parameters $\theta_i$ and new prompts $x_i$ to GENERATE
    asynchronously run TRAIN and GENERATE below

    **procedure** OFF-POLICY TRAIN$(x_{i-1}, y_{i-1})$    **procedure** GENERATE$(x_i, \theta_i)$
        reward samples $r_{i-1} \leftarrow R(x_{i-1}, y_{i-1})$        update generation model $\theta \leftarrow \theta_i$
        loss $l_{i-1} \leftarrow L(x_{i-1}, y_{i-1}, r_{i-1})$        generate new samples $y_i \sim \pi_{\theta_i}(x_i)$
        off-policy update $\theta_{i+1} \leftarrow \nabla_{\theta_i} l_{i-1}$

---

