# OpenReview forum: "Asynchronous RLHF: Faster and More Efficient Off-Policy RL for Language Models"
_ICLR.cc/2025/Conference — ICLR 2025 Poster_

### Official Review · Reviewer_rLXT · 2024-10-26

**Soundness:** 3
**Presentation:** 3
**Contribution:** 3
**Rating:** 6
**Confidence:** 4

**Summary:**

This paper studies incorporating off-policy asynchronous training to speed up RLHF. The main idea is to use separate GPUs for generation and training, which can leverage the acceleration techniques developed for both training and inference. Experiment results show that DPO is more robust under off-policy data, and the proposed method can achieve 40% speed up in training LLaMA 3.1 8B without sacrificing final performance.

**Strengths:**

This paper studies an important problem: how to speed up RLHF training. The proposed method is easy to implement and the experiment results are quite promising. I think it is quite meaningful to study such off-policy asynchronous training setting, as fully online training is quite expensive and leads to low utilization of GPUs. This paper gives a thorough study of such setting with meaning conclusions.

**Weaknesses:**

1. There are many other RLHF algorithms except PPO and DPO, such as IPO, KTO, etc. It would be great to see experiments on these algorithms.
2. It would be better to perform experiments on various types of dataset to further verify the results of this paper, as different types of dataset might have distinct properties. For example, it would be interesting to see the results on reasoning dataset.

**Questions:**

Nothing necessary stands out.

---

> ### Author Response · Authors · 2024-11-18
> **Rebuttal**
>
> We thank the reviewer for their comments and are glad our problem is important, our results are promising, and our conclusions are meaningful.
>
> **Other \*PO Methods**
>
> - PPO and Online DPO are current SOTA methods see [1](http://arxiv.org/abs/2404.10719) and [2](http://arxiv.org/abs/2402.04792)
> - IPO has shown very similar results to DPO in both [the original paper](http://arxiv.org/abs/2310.12036), and [follow-up work](http://arxiv.org/abs/2402.05749), we expect similar results here
> - [KTO](http://arxiv.org/abs/2402.01306) is a method for un-paired data. Our problems (summarization, general-purpose chatbots) generally have paired comparison data.
>
> We believe our results demonstrate that asynchronous versions of other *PO methods would also be faster and more efficient
>
> **Another Benchmark**
>
> On top of TLDR summarization and No Robots general-purpose chatbot, we are currently running experiments with GSM 8k as a reasoning / math benchmark. Thank you for the suggestion.

---

> > ### Author Response · Authors · 2024-11-26
> > **Novel Results**
> >
> > We have gone beyond the scope of original work and implemented the math and reasoning benchmark GSM8k. We achieve strong results with asynchronous training and thank the reviewer for the suggestion.
> >
> > In Appendix D.2 of the updated paper, we show how synchronous Online DPO improves on performance compared to an existing, highly-tuned PPO baseline. We then show how asynchronous Online DPO achieves the same final performance as synchronous with an even bigger speedup than any previous result, 68% faster. Overall, we outperform a state-of-the-art synchronous PPO baseline while running >6x faster on comparable hardware. See Table 8 on page 25.
> >
> > This is a big result and we plan on integrating it into the main results of the paper, replacing section 5.2 and extend our conclusion to generally RL for language models.
> >
> > Does the reviewer have any outstanding concerns?

---

> > > ### Comment · Reviewer_rLXT · 2024-11-30
> > > **response to rebuttal**
> > >
> > > Thanks for the thorough responses to my questions. I have no other conerns.

---

### Official Review · Reviewer_7CQN · 2024-10-28

**Soundness:** 3
**Presentation:** 4
**Contribution:** 3
**Rating:** 6
**Confidence:** 3

**Summary:**

This paper explores the possibility of asynchronous RLHF training for LLMs. Specifically, the generation and training steps, which constitute the majority of RLHF's processing time, are conducted asynchronously on separate GPUs. This approach naturally introduces staleness or so-called "off-policyness," meaning the training samples are generated by earlier versions of the LLM rather than the most recent one.

The authors find that off-policyness can significantly impact the performance of PPO across a wide range of model sizes. In contrast, DPO demonstrates greater resilience to off-policyness. Accordingly, the authors have built a system that employs online asynchronous DPO for RLHF, which can achieve 20% faster training than the synchronous approach in wall-clock time while delivering comparable or superior performance.

Currently, I don't find any critical flaws in this paper and would recommend for acceptance.

**Strengths:**

* This paper presents a novel perspective on off-policyness in RLHF. I especially appreciate the experiments comparing various RLHF loss functions and scaling behaviors related to off-policyness in Section 3. These experiments not only enrich this paper but also validate concurrent works on applying DPO to off-policy data and advocating for on-policy PPO training.

* The paper is well-organized and easy to follow.

**Weaknesses:**

While I don't have significant concerns that would prevent a positive recommendation, it would be helpful if the authors could clarify the following points to avoid potential misunderstandings.

## Motivation

The motivation presented in the first paragraph of Section 3 suggests that training libraries are slow for generation, and using separate libraries for training and generation could lead to GPU underutilization. This appears to be more of an implementation issue than a scientific one, which may not fully justify the exploration of off-policy RLHF. If RLHF systems can indeed achieve sufficient speed, investigating off-policy algorithms may not be necessary. For similar reasons, I have concerns regarding the experiments in Section 3.5.

## Experiments

The title "large-scale asynchronous RLHF" seems overstated. LLaMA-8b is not generally considered large-scale. A more accurate title could be "Asynchronous RLHF with LLaMA-8B."

Figure 9 should also include a comparison with online PPO and in more benchmark tasks. It's good to use TL;DR for analysis in Sec.3, but more diverse evaluation is required for the final experiment. The authors can consider benchmarks like SafeRLHF or MATH.

**Questions:**

+ How can we determine when the system is generation-bound or training-bound? Since generation length may vary during training, the system could be generation-bound at the start but training-bound later. The optimizations in Section 4 require selecting the paradigm in advance, which may not be feasible.

+ It was previously observed that off-policyness negatively affects learning performance in the OpenAI Five paper [1]. A dedicated section explaining this relationship could add clarity.

[1] Dota 2 with Large Scale Deep Reinforcement Learning, Figure 5, https://arxiv.org/pdf/1912.06680

---

> ### Author Response · Authors · 2024-11-18
> **Rebuttal**
>
> We thank the author for their review and appreciate that they recommend our work be accepted and find it well-organized, novel, and even enriching other concurrent work.
>
> **Why is efficient generation infeasible for synchronous RLHF**
>
> We thank the reviewer for the question. We answer below and have added a new Appendix C with a in-depth explanation.
>
> We use SOTA synchronous RLHF library, NVIDIA’s NeMo-Aligner, as an example. Training and Generation must use fundamentally different backends for efficiency (e.g. they use Megatron-LM for training, TensorRT-LLM for generation). NeMo-Aligner is an impressive engineering effort that integrates them both in synchronous RLHF but it adds overhead and still isn’t feasible to maintain because generation and training libraries are advancing too quickly in opposite directions.
>
> To make it work, NeMo-Aligner’s team wrote complex code to convert between training (Megatron-LM) and generation (TensorRT-LLM) backends. But maintaining this is difficult! Once the generation backend (TensorRT-LLM) was updated to v12 and then v13, it took the NeMO-Aligner team [1.5 months to integrate it into their library](https://github.com/NVIDIA/NeMo-Aligner/pull/320). By that time, TensorRT-LLM was updated again to a new version [v14](https://github.com/NVIDIA/TensorRT-LLM/releases/tag/v0.14.0) and NeMO-Aligner is again out-of-date. And this is a best-case scenario since all these libraries are from NVIDIA.
>
> Synchronous training libraries must consistently make big changes to integrate the latest generation code. Asynchronous training treats generation like an API and integrates new code easily.
>
> **Section 5 Title**
>
> We have changed the title of Section 5, thank you
>
> **PPO for 8B Chatbot**
>
> We are currently running synchronous and asynchronous PPO for our 8B general-purpose chatbot. Thank you for the suggestion.
>
> **Another Benchmark**
>
> On top of TLDR summarization and No Robots general-purpose chatbot, we are currently running experiments with GSM 8k as a reasoning / math benchmark. Thank you for the suggestion.
>
> **Training vs Generation-Bound**
>
> In order to account for compute changes during training, it is possible to make best-of-$K$ samping and num ppo epochs $N$ into hyperparameters that anneal / change with time or compute considerations. In practice, we didn't find significant changes during our training.
>
> **Off-Policy Degradation in OpenAI Five**
>
> This is a great pointer, thank you. We’ve added this citation and note to the end of Section 3.2

---

> > ### Comment · Reviewer_7CQN · 2024-11-20
> > **Reply by by Reviewer 7CQN**
> >
> > I appreciate the authors' response but still have two remaining concerns:
> >
> > 1. **Determining Whether the Off-Policy Paradigm is Training or Generation-Bound**
> >
> > I am uncertain how to determine whether the paradigm is training-bound or generation-bound before launching the experiment. While it is possible to adjust hyperparameters to address discrepancies during training, how should one set an appropriate initial value?
> >
> > 2. **Analysis of Synchronous RLHF Efficiency**
> >
> > Thank you for the detailed analysis of why implementing efficient synchronous RLHF is challenging. However, I have the following concerns:
> >
> > * According to recent research [1], NeMo-Aligner does not appear to be the SOTA system. In fact, it is reportedly slower than another synchronous system, DeepSpeed-Chat. Neither of these systems was used in the experiments of this paper. If the authors aim to demonstrate that asynchronous training is faster and more efficient, could you provide an end-to-end comparison? For example, comparing DeepSpeed-Chat or NeMo-Aligner with the proposed system would strengthen the argument.
> >
> > * The analysis seems to focus on a specific issue within NeMo-Aligner or NVIDIA's implementation. It is not sufficiently substantiated to claim that this problem would occur in all future asynchronous libraries, such as those described in [1] and [2]. This appears to be more of an implementation or software engineering limitation rather than a fundamental scientific issue, which weakens the paper's motivation.
> >
> > [1] HybridFlow: A Flexible and Efficient RLHF Framework, https://arxiv.org/abs/2409.19256
> >
> > [2] ReaLHF: Optimized RLHF Training for Large Language Models through Parameter Reallocation, https://arxiv.org/abs/2406.14088

---

> ### Author Response · Authors · 2024-11-21
> **Rebuttal and Clarification**
>
> Thank you for your quick response and deep dive into our problem
>
> **1. Determining training/generation-bound**
>
> We keep things simple and launch a short synchronous test experiment to get an idea of training/generation speeds per iteration. We then determine how many GPUs to use and divide our GPUs using the ratio of training/generation speeds.
>
> **2a. Our research contribution vs engineering work (NeMo-Aligner)**
>
> We’d like to clarify a major point: **we are not proposing an engineering solution but a research paradigm (asynchronous + off-policy learning)**. Our contribution is not directly comparable to engineering works such as NeMo-Aligner or HybridFlow. These libraries show how to optimize allocations and pipelines in the existing paradigm, but they don’t suggest new paradigms, they don’t show learning curves, or convergence. They focus purely on throughput of tokens. E.g. [NeMo-Tensor’s LLaMA-3 model trained with PPO is worse than the LLaMA-3 instruct model](https://huggingface.co/nvidia/Llama3-70B-PPO-Chat#evaluation) and HybridFlow doesn’t show a single learning curve or demonstrate any results that its pipeline can effectively train a model.
>
> In contrast, we demonstrate that asynchronous learning is a new paradigm that can achieve equal performance more efficiently, an alternative to the synchronous status quo. We focus on the effectiveness of the trained model. Since this paradigm is novel, there does not exist a fully optimized library for asynchronous RLHF for comparison, and creating it would be a big engineering effort.
>
> **2b. Our Motivation is General: HybridFlow is already out of date**
>
> Integrating updates is not just an issue for NeMo-Tensor, but all synchronous RLHF libraries that aim to integrate training and generation libraries into one. The reviewer’s suggestion, HybridFlow, has it much worse than NeMo-Tensor. Despite being released 3 weeks ago, it uses [Megatron 0.4.0 and vllm 0.5.4](https://github.com/volcengine/verl?tab=readme-ov-file#dependencies). These libraries are, respectively, [11 months](https://github.com/NVIDIA/Megatron-LM/releases/tag/core_v0.4.0) and [3.5 months](https://github.com/vllm-project/vllm/releases/tag/v0.5.4) out of date. HybridFlow, despite being just released, does not support [MoE](https://github.com/vllm-project/vllm/releases/tag/v0.5.5), LLaMA 3.2, or [multistep scheduling](https://github.com/vllm-project/vllm/releases/tag/v0.6.0) which 2x improves generation throughput.
>
> We believe our motivation is a fundamental limitation: not fundamental scientific possibility but fundamental *feasibility*. Whereas synchronous libraries have complex combinations of tricks to co-locate training and generation, our proposed asynchronous paradigm is effective while being drastically simpler.

---

> > ### Comment · Reviewer_7CQN · 2024-11-22
> > **Thank you for the further clarification.**
> >
> > Thank you for the further clarification. While my first concern has been addressed, I still have some comments regarding the second concern.
> >
> > The authors claim to propose a research paradigm (i.e., asynchronous learning) for LLM RLHF. However, the contribution of this point appears weak because asynchronous learning is already a well-adopted and well-tested paradigm in large-scale RL, as demonstrated in applications like Dota2. The primary contribution of this paper, instead, lies in showing the feasibility (or infeasibility) of this paradigm and providing corresponding analyses under specific circumstances. While I agree that this is not simply an engineering solution, I am unclear why a direct comparison with NeMo-Aligner or DS-chat is considered invalid (though comparing with HybridFlow is indeed invalid since it was released after the ICLR submission deadline). The algorithmic implementation is orthogonal to what the authors describe as "engineering efforts." A minimal modification, such as adapting your PPO code to a baseline synchronous system, could enable an end-to-end comparison. Even if the learning curves of the baseline systems are not explicitly reported in their papers, they could still serve as a reference for comparison. Only when the proposed asynchronous paradigm is compared with **an optimized synchronous solution** can we truly evaluate whether the asynchronous approach offers advantages over existing methods.
> >
> > I appreciate the authors’ detailed explanation of why HybridFlow is already outdated, and I fully agree with this assessment. I now understand that implementation issues indeed exist in many current systems. However, this argument hinges on the assumption that the synchronous system targets distributed training, rather than operating on a single node. Therefore, if the authors intend to use this fact to support the claim that the asynchronous paradigm has a fundamental advantage, the paper needs experiments to demonstrate that **asynchronous approaches outperform synchronous ones in large-scale distributed scenarios**. Without such experiments, the implementation-wise comparison (i.e., Appendix C and the authors' rebuttal) does not serve as valid evidence.

---

> > > ### Author Response · Authors · 2024-11-26
> > >
> > > We’re glad to have cleared up one concern and appreciate your commitment to continued discussion, it is helping us improve our work even further. We also have novel results to share.
> > >
> > > **This work is about the viability and promise of Asynchronous RLHF**
> > >
> > > We believe that our asynchronous RLHF setup is slower than current state-of-the-art synchronous RLHF libraries: changing from a dominant paradigm *and* outperforming well-tuned methods from the previous paradigm is a large task that requires a research agenda spanning many works (showing viability, optimizing losses, optimizing distributed computation etc).. A whole line of research and [many iterations of libraries](https://github.com/opendilab/awesome-RLHF?tab=readme-ov-file#codebases) have already gone into optimizing synchronous RLHF for at least 4 years. But to avoid getting stuck in the synchronous paradigm due to a [hardware lottery](https://arxiv.org/abs/2009.06489), we have demonstrated the viability and advantages of asynchronous RLHF.
> > >
> > > Our work is just the beginning of the research agenda. Asynchronous RLHF is novel because it was not clear that off-policyness could be tolerated in RLHF training. By demonstrating promising results, we lay the groundwork so the community has a reason to try optimizing this paradigm. We have softened the conclusion of our paper in line with this idea, thank you.
> > >
> > > **Large-Scale Distributed is Out of Scope**
> > >
> > > We think this is beyond the scope of our current work. The engineering and compute resources required to make asynchronous RLHF work across large-scale distributed systems constitutes its own research project, at minimum. NeMo-Tensor accomplishes this for synchronous RLHF in 17000 lines of python code, with [13 different contributors that each added at least 1000 lines of code](https://github.com/NVIDIA/NeMo-Aligner/graphs/contributors), a serious endeavour. We do not know whether asynchronous RLHF will be better in large-scale systems, but by publishing our work, we hope to encourage teams with such large resources to find out.
> > >
> > > **Novel Result: Sync vs Async PPO experiments**
> > >
> > > We have run Sync vs Async PPO for our 8B chatbot, please see the new Appendix D.1. We find that, once again, async training matches sync while being faster. Online DPO is still more performant than PPO for this task. Thank you for this suggestion, we believe it strengthens our contribution.
> > >
> > > **Novel Result: Asynchronous RL for Math/Reasoning on GSM8k**
> > >
> > > We have gone beyond the scope of original work and achieved strong results with asynchronous training on the math and reasoning benchmark GSM8k. In Appendix D.2 we show how synchronous Online DPO improves on performance compared to an existing, highly-tuned PPO baseline. We then show how asynchronous Online DPO achieves the same final performance as synchronous with an even bigger speedup than any previous result, 68% faster. Overall, we outperform a state-of-the-art synchronous PPO baseline while running 6x faster on comparable hardware.
> > >
> > > This is a big result and we thank the reviewer for suggesting it. We believe we should integrate it into the main results of the paper, replacing section 5.2 and extend our conclusion to generally RL for language models. We are open to feedback and suggestions.

---

> > > > ### Comment · Reviewer_7CQN · 2024-11-26
> > > > **Response by Reviewer 7CQN**
> > > >
> > > > I would like to express my sincere gratitude to the authors for their in-depth discussion with me. Their responses have effectively addressed my concerns.
> > > >
> > > > After reviewing comments from other reviewers, I believe there are no major weaknesses that would justify rejecting this paper. While I will not change my score, I will advocate for its acceptance during discussions with my fellow reviewers.
> > > >
> > > > Thank you again to the authors for their excellent paper and insightful discussion.

---

> > > > > ### Author Response · Authors · 2024-12-03
> > > > >
> > > > > We'd also like to thank the reviewer for their time and effort in this discussion. It has been deeply helpful and we really appreciate the dedication to understand the details of our work and help us improve it.
> > > > >
> > > > > Since the score of 6 is "borderline accept", we feel the reviewer may consider increasing their score as their attitude seems more in line with a standard "accept". Regardless, we are glad for the discussion and grateful to the offer of advocacy.

---

### Official Review · Reviewer_6pCS · 2024-11-02

**Soundness:** 2
**Presentation:** 2
**Contribution:** 2
**Rating:** 6
**Confidence:** 3

**Summary:**

This paper discusses the possibility to achieve faster and more efficient RLHF training of LLMs by exploiting online but off-policy data. The authors propose to optimize compute efficiency by separating the generation and training tasks in RLHF on different sets of GPUs and executing them asynchronously. However, this requires training on off-policy samples. This paper further explores properties of asynchronous off-policy RLHF on TLDR Summarization benchmark. The results show that the online DPO method is the most robust to off-policy data, and robustness increases with the scale of the policy model. Moreover, some optimization methods targeting generation-bound and training-bound scenarios are also proposed to further increase the efficiency of asynchronous RLHF. Finally, the authors train a LLaMA 3.1 8B model on an instruction following task with online DPO, showing that the asynchronous method is 40% faster than its synchronous counterpart.

**Strengths:**

1. The paper is well-organized and the ideas could be easily understood.
2. The asynchronous training in RLHF of LLMs is an relatively unexplored topic compared to synchronous methods. This paper has made a first attempt to research in this area.
3. The experiments on the robustness of different RLHF algorithms to off-policy data is abundant. It could be inspiring to researches in related topics.

**Weaknesses:**

1. For asynchronous RLHF, it is unclear to me where the computation efficiency increase comes from. Compared to synchronous RLHF, does asynchronous RLHF reduce communication overheads or GPU idle time? Or does it decrease the overall computation required to achieve the same goal? It would be much more easier for readers to understand the performance gain if the authors could provide a breakdown analysis in the experiments.
2. From the first paragraph in Section 3, it seems that a major obstacle that prevents the synchronous RLHF to be efficient is that there are no efficient implementation of generation in existing training libraries. From my perspective, this is only an issue that could be solved by engineering efforts. If this issue is solved, what is the advantage of asynchronous RLHF? Alternatively, could you explain why it is infeasible to make generation efficient in the training libraries?
3. The authors have also mentioned in the footnote that there are a more advanced approach to interleave training and generation [1]. However, besides the paged memory in vLLM (which will reserve a large portion of GPU memory for KV cache), there are also many optimizations that could be applied to the generation (such as CUDAGraph, speculative decoding, continuous batching, etc.) in RLHF without harming the performance of training tasks, making it much more faster than the huggingface transformers implementation. If these optimization methods are applied, what is the advantage of asynchronous RLHF compared to the approach in [1]?
4. In section 5, the experiment result only shows the asynchronous version of online DPO compared to the synchronized version of online DPO. However, it is still unclear how the asynchronized online DPO performs compared to other synchronized algorithms with the same amount of computational resources. Could you provide cases in which the asynchronized DPO outperforms the synchronized version of another RLHF algorithm (e.g. PPO)? This would make the advantage of asynchronous RLHF much more convincing.
5. The benchmark used in this paper is TLDR Summarization, which is relatively simple compared to problems such as coding and math.  As an extension to problem 4, will the asynchronized online DPO perform better than the synchronized PPO in these harder problems?

[1] Mei, Z., Fu, W., Li, K., Wang, G., Zhang, H., & Wu, Y. (2024). ReaLHF: Optimized RLHF Training for Large Language Models through Parameter Reallocation. https://arxiv.org/abs/2406.14088

**Questions:**

Please respond to the questions in the weaknesses section.

---

> ### Author Response · Authors · 2024-11-18
> **Rebuttal and Update**
>
> We thank the reviewer for their in-depth assessment and critiques, we appreciate that they find our paper well-organized, novel, and inspiring.
>
> Here, we try to clarify the why of asynchronous learning and we’ve added a new Appendix C with a more in-depth explanation.
>
> **1. Why is async more efficient**
>
> Our speed improvements come from SOTA generation libraries, which have separate optimizations from training. In Appendix C.1 we show how much faster vllm is than huggingface transformers.
>
> Synchronous RLHF is slower since either (1) it uses training libraries for generation, which makes generation really slow or (2) it uses SOTA generation on separate GPUs, but incurs GPU idle time: training GPUs are idle while waiting for generation GPUs to finish and vice versa. Asynchronous RLHF uses SOTA generation on separate GPUs and reduces GPU idle time by parallelizing generation and training.
>
> **2. Why is efficient generation infeasible for synchronous RLHF**
>
> Syncronous RLHF can also (3) interleave SOTA generation and training on the same GPUs. This is actually available in NVIDIA’s SOTA RLHF library, NeMo-Aligner, an impressive engineering effort. But it adds communication overhead and we argue it isn’t feasible to maintain as generation and training libraries are advancing too quickly in opposite directions.
>
> To make it work, NeMo-Aligner’s team wrote complex code to convert between the separate training (Megatron-LM) and generation (TensorRT-LLM) libraries. But maintaining this is difficult! Once the generation library (TensorRT-LLM) was updated to v12 and then v13, it took the NeMO-Aligner team [1.5 months to integrate it into their library](https://github.com/NVIDIA/NeMo-Aligner/pull/320). By that time, TensorRT-LLM was updated again to a new version [v14](https://github.com/NVIDIA/TensorRT-LLM/releases/tag/v0.14.0) and NeMO-Aligner is again out-of-date. And this is a best-case scenario since all these libraries are from NVIDIA.
>
> Synchronous training libraries must consistently make big changes to integrate the latest generation code. Asynchronous training treats generation like an API and integrates new code easily. Thank you for this question and please see our new Appendix C.2 for a more detailed explanation.
>
> **3. Why can’t training use generation optimizations**
>
> Many newer optimizations are incompatible with training (e.g. paged-attention). Other generation tricks can be integrated in synchronous training but you usually pay a cost. E.g. NeMo-Aligner compiles the model into a CUDA graph but this takes up memory on the GPU. This means less available memory during training so NeMo-Aligner has to compensate with even more tricks to reduce optimizer memory (gradient accumulation and gradient checkpointing) but these make training slower.
>
> **4. Sync vs Async PPO**
>
> We do not necessarily recommend async Online DPO, but rather async vs sync. We are currently running sync and async PPO for our general-purpose 8B chatbot to show that async PPO can match sync PPO while being faster. Thank you for the suggestion.
>
> **5. Another benchmark**
>
> On top of TLDR summarization and No Robots general-purpose chatbot, we are currently running experiments with GSM 8k as a reasoning / math benchmark. Thank you for the suggestion.

---

> ### Comment · Reviewer_6pCS · 2024-11-20
>
> Thank you for the detailed responses and the addtional appendix. However, a few concerns still remain:
>
> 1. In Appendix C.2., you have mentioned the overheads in dynamic model resharding. It would be more convincing if you could give a small example on how much the overhead is in a typical training experiment using NeMo-Aligner.  Specifically, how much percentage of time does it take to reshard parameters in one training iteration?
>
> 2. You have mentioned that NeMo-Aligner's team wrote complex code to convert training and generation libraries, and it is difficult to maintain the code when the libraries update. This seems a specific engineering flaw in the implementation of NVIDIA's training and generation library. I understand that the authors may not know about the underlying details of the development of NeMo-Aligner and related NVIDIA's libraries, but it seems that an important argument in the paper relies on this specific engineering flaw and it is not well-discussed.
>
> 3. About CUDAGraph, it does take extra memory on the GPUs and could not be freed during training in NeMo-Aligner. However, it is not necessary since CUDAGraph could be destroyed before training and recaptured before the next generation. In this way, the memory space for CUDAGraph will be freed when training. I believe that this is another engineering flaw in NeMo-Aligner.
>
> To show that your argument is not solely established on the engineering flaws of NeMo-Aligner, I recommend that you could  provide an end-to-end comparison between your asynchronous implementation and synchronous open-source libraries including NeMo-Aligner and others (e.g. DSChat).

---

> ### Author Response · Authors · 2024-11-21
> **Rebuttal and Clarifications**
>
> Thank you for the fast response and reading our appendix, we appreciate this discussion.
>
> **1. Resharding Overhead**
>
> We are not experts in NeMo-Aligner, so we can’t give exact estimates and overhead as a percentage depends on the number of tokens generated for responses, number of PPO epochs etc.. Since this resharding is a dynamic design decision mentioned by the authors, we believe it is time-consuming enough to matter.
>
> **2. Training and Generation must use different backends**
>
> Every state-of-the-art RLHF library uses different backends for training and generation to better optimize each. Training is a dynamic process which focuses on faster backprop and generation is a fixed process that focuses on faster throughput.
> - [RealHF](https://arxiv.org/pdf/2406.14088) (Megatron-LM/Deepspeed and vLLM)
> - NeMo-Tensor (Megatron-LM, TensorRT-LLM)
> - [OpenRLHF](https://github.com/OpenRLHF/OpenRLHF) (Deepspeed, vLLM)
>
> It is not a flaw, but an achievement that the backends are different: it shows how much generation has been optimized in ways that don’t apply to training. RLHF libraries that try to use one backend are too slow e.g. [OpenRLHF](https://arxiv.org/abs/2405.11143) notes “the generation acceleration of vLLM is substantially better than that of [Deepspeed-Chat] Hybrid Engine”.
>
> We hope this is convincing that we are motivated by a fundamental limitation of all synchronous RLHF libraries
>
> **3. CUDA graph / engine cannot be freed and recreated**
>
> Building the CUDA graph is very computationally expensive and prohibitive to do each iteration. From NeMo-Aligner, “due to the cost of serializing the engine, we keep the engine in memory during training” and “the first step incurs substantial time for TensorRT-LLM Engine Building”
>
> **Our research contribution vs engineering work**
>
> We’d like to clarify a major point: **we are not proposing an engineering solution but a research paradigm (asynchronous + off-policy learning)**. Our contribution is not directly comparable to NeMo-Aligner, an engineering work. Engineering works optimize allocations and pipelines in an existing paradigm and focus purely on throughput of tokens. They don’t suggest new paradigms, show learning curves, or look at convergence.. E.g. [NeMo-Tensor’s LLaMA-3 model trained with PPO is worse than the LLaMA-3 instruct model](https://huggingface.co/nvidia/Llama3-70B-PPO-Chat#evaluation). Comparison to NeMo-Aligner was also not possible at submission time since, as noted in Appendix C.3, it did not support LLaMA 3.1
>
> In contrast, we demonstrate that asynchronous learning is a new paradigm that can train models with equal performance but more efficiently, an alternative to the synchronous status quo. We focus on the effectiveness of the trained model. Since this paradigm is novel, there does not exist a fully optimized library for asynchronous RLHF, which would be a big engineering effort.

---

> > ### Comment · Reviewer_6pCS · 2024-11-22
> >
> > I appreciate the authors for the clarification! Meanwhile, there are still some points I need to mention:
> >
> > 1. I understand you are mainly proposing the asynchronous research paradigm in this paper. However, an important statement in your paper is that **asynchronous RLHF is more computationally efficient than its synchronous counterpart**. To support this statement, it is crucial for you to show that your approach is able to outperform SOTA synchronous RLHF systems. Otherwise your statement should be a lot weaker: **asynchronous RLHF can achieve a high computational efficiency with less engineering effort**.
> >
> > 2. It is true that it is not very reasonable to require the authors to study the details of NeMo-Aligner and conduct a performance breakdown. However, I still believe that this paper lacks an in-depth analysis to why asynchronous RLHF is more efficient. Other than an experiment involving other baselines, you could also theoretically analyse or perform a breakdown analysis in your own implementation to show how much overheads your asynchronous approach eliminates compared to SOTA synchronous approaches.
> >
> > 3. When I mentioned that its a flaw of NeMo-Aligner that it is difficult to maintain the codes converting training and generation libraries, I did not imply that it is bad to use different training and generation backends. By mentioning this, I intended to ask following questions:
> >     - Does this "difficult-to-maintain" scenario **only** happen in NVIDIA's libraries? Or is it a **general** problem for all existing libraries?
> >     - If it is a general problem, why?
> >
> > I believe it could improve the paper a lot if the authors could answer the above-mentioned questions. However, I still agree that the asynchronous RLHF paradigm is a very promising research topic and this paper is beneficial to the community. Therefore I am willing to improve my rating by 1.

---

> ### Author Response · Authors · 2024-11-26
>
> Thank you for your notes and continued discussion, it is helping us improve our work even further. We also have novel results to share.
>
> **1. This work is about the viability and promise of Asynchronous RLHF**
>
> We agree that our asynchronous RLHF is slower than current state-of-the-art synchronous RLHF libraries: changing a paradigm *and* outperforming previous SOTA is a large task, arguably a whole research agenda. Lots of effort has gone into optimizing synchronous RLHF which makes it hard to immediately compete with. But to avoid getting stuck in a [hardware lottery](https://arxiv.org/abs/2009.06489), we have demonstrated the viability and advantages of asynchronous RLHF. Our work is the beginning of the research agenda, laying the groundwork so the community can investigate and begin to optimize this different paradigm.
>
> We have softened the conclusion of our paper in line with this idea, thank you.
>
> **2. Why Asynchronous is more efficient**
>
> In Appendix C.1 we demonstrate the need for generation libraries. In section 5.2 and at the end of the new Appendix D (on the final page), we examine the exact runtimes and how asynchronous learning speeds up training over a simple, but competitive, synchronous RLHF by reducing idling time.
>
> **3. Difficult maintenance is a fact of all synchronous, integrated RLHF libraries**
>
> Synchronous RLHF libraries manually integrate training and generation libraries to work around synchronous limitations (constant switching between train and generate). Updates generally break those manual integrations.
>
> NeMo-Aligner is not just an example but a best case scenario. It is maintained by a team of strong engineers and the underlying training and generation frameworks are from the same company. For an average case, [HybridFlow](https://arxiv.org/abs/2409.19256) was released 3 weeks ago, it uses [Megatron 0.4.0 and vllm 0.5.4](https://github.com/volcengine/verl?tab=readme-ov-file#dependencies). These libraries are, respectively, [11 months](https://github.com/NVIDIA/Megatron-LM/releases/tag/core_v0.4.0) and [3.5 months](https://github.com/vllm-project/vllm/releases/tag/v0.5.4) out of date. HybridFlow, despite being just released, does not support [MoE](https://github.com/vllm-project/vllm/releases/tag/v0.5.5), LLaMA 3.2, or [multistep scheduling](https://github.com/vllm-project/vllm/releases/tag/v0.6.0) which 2x improves generation throughput.
>
>
>
> **Novel Result: Sync vs Async PPO experiments**
>
> We have run Sync vs Async PPO for our 8B chatbot, please see the new Appendix D.1. We find that, once again, async training matches sync while being faster. Online DPO is still more performant than PPO for this task. Thank you for this suggestion, we believe it strengthens our contribution.
>
> **Novel Result: Asynchronous RL for Math/Reasoning on GSM8k**
>
> We have gone beyond the scope of original work and achieved strong results with asynchronous training on the math and reasoning benchmark GSM8k. In Appendix D.2 we show how synchronous Online DPO improves on performance compared to an existing, highly-tuned PPO baseline. We then show how asynchronous Online DPO achieves the same final performance as synchronous with an even bigger speedup than any previous result, 68% faster. Overall, we outperform a state-of-the-art synchronous PPO baseline while running 6x faster on comparable hardware.
>
> This is a big result and we thank the reviewer for suggesting it. We believe we should integrate it into the main results of the paper, replacing section 5.2 and extend our conclusion to generally RL for language models. We are open to feedback and suggestions.

---

### Official Review · Reviewer_dsGt · 2024-11-03

**Soundness:** 2
**Presentation:** 2
**Contribution:** 2
**Rating:** 5
**Confidence:** 3

**Summary:**

The paper propose a asynchronous approach for accelerating the training of RLHF when multiple GPUs are available. Specifically, they propose to do generation and training simultaneously by letting policy learn on off-policy data. Experiments on TLDR dataset shows that the algorithm can acclerate the training by about 40%. Additionally, the paper provides some empirical results on the effect of how the off-policy data may affect the policy learning.

**Strengths:**

* The asynchronous implementation enables model to do generation and training simultaneously and reduces the time for waiting reward signal. Experiments in finetuning Llama 3.1 shows a 40% acceleration in terms of reward convergence.

**Weaknesses:**

* The study on the effect of using off-policy data for updating is not very convincing. Based on the protocol in the paper (line 200-207), the policy seems to be trained on the data generated by the *same* outdated policy, which may be the key that results in performance drop. Specifically, the data generated by the same policy may be homogeneous (think about an extreme case where the policy generates $N$ same data pairs). In this case, the model only needs to learn a small update to adapt on these data, resulting a small KL divergence. A better experimental design is to train model using data generated by *different model* (that is $N$ steps behind) for each update.

* Given different computational facility and tasks, it is not trivial to design an optimal resource allocation scheme for generation and training, which may require heavy manual tuning.

**Questions:**

Please refer to my weakness part.

---

> ### Author Response · Authors · 2024-11-18
> **Rebuttal**
>
> We thank the reviewer for their comments and we’re glad they appreciate our improvements in speed and efficiency. We agree this is a key part of our paradigm.
>
> **Off-Policy Setup: Same vs Different Policy?**
>
> Our setup is made to mimic how asynchronous learning requires off-policy learning. For the off-policy setup, we already sample from our own policy. And as we update, our final sample is from our own policy, now $N$ steps behind. Empirically, we don’t see any issue of homogenous data since we sample with temperature 1 and there is a consistent difference in reward between our two responses per prompt.
>
> [Other work](http://arxiv.org/abs/2405.08448) has already noted that off-policyness should reduce performance. Can the reviewer explain which exact model is being suggested for sampling?
>
> **Resource Allocation**
>
> We agree that designing an optimal resource allocation strategy can be difficult. However, our tuning is straightforward: if generation is slower then use more GPUs for generation, and if training is slower we can run more GPUs for training. And synchronous RLHF libraries like [NeMO-Aligner](https://arxiv.org/abs/2405.01481) already deal with this because they use asynchronous reward models and have existing ways for reward and policy compute allocation.
>
> We thank the reviewer for the comment and have added a new section in Appendix C.3 to discuss the issue of resource allocation. Does this alleviate the concern?

---

> > ### Author Response · Authors · 2024-11-26
> > **Novel Results**
> >
> > Does the reviewer have any outstanding concerns preventing them from recommending acceptance? The discussion period is ending soon and we believe we have addressed all issues.
> >
> > On other reviewers' suggestion, we have gone beyond the scope of original work and implemented the math and reasoning benchmark GSM8k. We achieve strong results with asynchronous training and wanted to note these new, strong results in Appendix D.2 of the updated paper. Overall, we show an asynchronous Online DPO outperform a state-of-the-art synchronous PPO baseline while running >6x faster on comparable hardware. See Table 8 on page 25.

---

> > ### Comment · Reviewer_dsGt · 2024-11-26
> > **Further questions regarding the off-policy update**
> >
> > Thanks for the response.
> >
> > Based on my understanding, if the generation is faster than training, then the proposed method uses the reward that is only one-step behind to train the model. If the generation is slower than training, the model needs to wait the generation. To leverage this waiting time, the model will be trained on the past rewards.
> >
> > So my question is, for the second case, how does the model determine which past rewards are selected for update? If it is in the same scenario as the Section 3.1 (as indicated in authors' last response), which is to update for $N$ times based on $N$ batch samples generated by the last policy. Then, after this round of update, what will the next data batch come from?
> >
> > I would expect authors to provide a clear explanation on off-policy data's utilization.

---

> > > ### Author Response · Authors · 2024-11-26
> > > **Clarification**
> > >
> > > Thank you for your response. Your understanding isn't exactly correct.
> > >
> > > Regardless of training-bound or generation-bound, asynchronous RLHF reduces idling time. In synchronous RLHF, training has to wait for generation to finish and then the generation has to wait for training to finish. Asynchronous RLHF removes the waiting by running both at the same time, generating new completions $y_i \gets \pi_{\theta_i}(x_i)$ and training off-policy on samples from the previous step $\nabla_{\theta_i} R(x_{i-1},y_{i-1})$.
> > >
> > > So for the next batch of prompts, we generate new completions $y_{i+1} \gets \pi_{\theta_{i+1}}(x_{i+1})$ and train on the ones we generated in the last step $\nabla_{\theta_{i+1}} R(x_{i},y_{i})$. For a visual comparison of simple, efficient synchronous training and asynchronous, see Figure 14 on the last page of the Appendix. We've also added a clear algorithm pseudocode on the same page in Appendix E.
> > >
> > > This is for $N=1$ mini-batches, and therefore $N=1$ steps off-policy. We can generalize it to $N>1$ by making $x_i$ contain $N$ times more data and so taking $N$ times more training steps between generations. But for our large scale experiments, we find $N=1$ to be sufficient.
> > >
> > > Does this clarify things?

---

> ### Comment · Reviewer_dsGt · 2024-11-29
>
> Thanks for the response, but my concern is not well addressed.
>
> Specifically, consider the scenario where generation is slower than training. After training model for the first $N$ steps ($N > 1$), there is less than $N$ samples generated. Then, where does the next data batch come from? Could authors explain the detailed protocol? These information are not included in Algorithm.

---

> ### Author Response · Authors · 2024-11-30
> **Further clarification**
>
> Our [Cleanba (Huang et al, 2023)](https://arxiv.org/abs/2310.00036) style asynchronous learning is not *completely* asynchronous. Training and generation sync every $N$ steps (e.g. $N=1$ in our general-purpose chatbot experiments in Section 5). So if generation is slower, training will idle for generation to finish (and vice-versa). We will make this detail clearer in the Algorithm in the Appendix.
>
> As shown in Figure 6, training will wait for generation to finish in order to get the next batch of data. Asynchronous RLHF does not eliminate idling, but can reduce it greatly.

---

> > ### Author Response · Authors · 2024-12-03
> >
> > The discussion period is now over. Thank you for the interactions!
> >
> > If we've addressed your final concern, please consider increasing your score.
> >
> > We'd also like to point out our new strong results on reasoning + RL, where asynchronous training is 68% faster, even more than for RLHF.

---

### Author Response · Authors · 2024-12-03
**Discussion Period Summary**

After useful and fruitful discussions with all reviewers, we seem to have assuaged all concerns
- reviewer 7CQN has no remaining concerns and recommends acceptance noting it is an "excellent paper" and they "will advocate for its acceptance"
- reviewer rLXT has no remaining concerns and recommends acceptance noting it is an "important problem" and "this paper gives a thorough study of [asynchronous RLHF] with meaning[ful] conclusions."
- reviewer 6pCS had a few remaining points that we clarified and recommends acceptance noting it "is a very promising research topic and this paper is beneficial to the community"
- reviewer dsGt had one last minor point to clarify and we hope we have adequately addressed it

We thank all reviewers for their time and effort and note the biggest changes from the discussion period.

**improved motivation**

Thanks to deep discussions with 6pCS and 7CQN, we have greatly strengthened our motivation and added a substantial appendix C describing the engineering details and a case study for the challenges of synchronous RLHF. We go into detail why asynchronous RLHF is a simpler, effective approach. We thank the reviewers for helping clarify our work.

**strong new results in reasoning / math**

Thanks to a suggestion by 6pCS, 7CQN, and rLXT, we have also gone beyond the scope of our initial paper and achieved asynchronous RL with LLMs for math/reasoning. In a new appendix D.2, we compare to a state-of-the-art PPO baseline using the standard GSM8k dataset. Our synchronous Online DPO setup runs nearly 4x faster and achieves better final results than PPO. Our asynchronous Online DPO is comparable and even slightly better than synchronous, while running 68% faster. This speedup is even larger than our previous results and demonstrates the promise of asynchronous RL for LLMs. We thank the reviewers for the suggestion and we plan to incorporate this very strong result into the main paper.

---

### Meta-Review · Area_Chair_vAop · 2024-12-23

**Metareview:**

This paper discusses the possibility to achieve faster and more efficient RLHF training of LLMs by exploiting online but off-policy data. The authors propose to optimize compute efficiency by separating the generation and training tasks in RLHF on different sets of GPUs and executing them asynchronously. However, this requires training on off-policy samples. This paper further explores properties of asynchronous off-policy RLHF on TLDR Summarization benchmark. The results show that the online DPO method is the most robust to off-policy data, and robustness increases with the scale of the policy model. Moreover, some optimization methods targeting generation-bound and training-bound scenarios are also proposed to further increase the efficiency of asynchronous RLHF. Finally, the authors train a LLaMA 3.1 8B model on an instruction following task with online DPO, showing that the asynchronous method is 40% faster than its synchronous counterpart.

All reviewers and the AC agree that this paper studies an emerging and important problem and this paper makes non-trivial technical contributions. The AC thus recommends acceptance.

**Additional Comments On Reviewer Discussion:**

There were concerns about the use of off-policy data. The AC believes the authors gave convincing responses and thus recommends acceptance.

---

### Decision · Program_Chairs · 2025-01-22

Accept (Poster)